# Redox driven B$_{12}$-ligand switch drives CarH photoresponse

Harshwardhan Poddar [1], Ronald Rios-Santacruz [2], Derren J. Heyes [1], Muralidharan Shanmugam [3], Adam Brookfield[3], Linus O. Johannissen [1], Colin W. Levy[1], Laura N. Jeffreys [1], Shaowei Zhang[1], Michiyo Sakuma[1], Jacques-Philippe Colletier[2], Sam Hay [1], Giorgio Schirò[2], Martin Weik[2], Nigel S. Scrutton [1] & David Leys [1]

CarH is a coenzyme B$_{12}$-dependent photoreceptor involved in regulating carotenoid biosynthesis. How light-triggered cleavage of the B$_{12}$ Co-C bond culminates in CarH tetramer dissociation to initiate transcription remains unclear. Here, a series of crystal structures of the CarH B$_{12}$-binding domain after illumination suggest formation of unforeseen intermediate states prior to tetramer dissociation. Unexpectedly, in the absence of oxygen, Co-C bond cleavage is followed by reorientation of the corrin ring and a switch from a lower to upper histidine-Co ligation, corresponding to a pentacoordinate state. Under aerobic conditions, rapid flash-cooling of crystals prior to deterioration upon illumination confirm a similar B$_{12}$-ligand switch occurs. Removal of the upper His-ligating residue prevents monomer formation upon illumination. Combined with detailed solution spectroscopy and computational studies, these data demonstrate the CarH photoresponse integrates B$_{12}$ photo- and redox-chemistry to drive large-scale conformational changes through stepwise Co-ligation changes.

CarH proteins are a unique class of coenzyme B$_{12}$ or adenosylcobalamin (AdoCbl) binding photoreceptors involved in the regulation of the carotenoid biosynthetic genes in several bacteria[1–11]. AdoCbl-bound CarH forms a tetramer in the dark, consisting of a dimer of dimers through interactions between cobalamin-binding domains (CBD) (Fig. 1a, b). The tetramer can suppress transcription by binding to a specific promoter region via the N-terminal DNA-binding domains (DBD) (Fig. 1c)[12]. In the dark-state tetramer, each monomer can bind a light-sensitive AdoCbl chromophore (Fig. 1a, e), although tetramer formation does not require a full chromophore complement[13]. Upon illumination, the protein in solution dissociates into light-adapted monomers, concomitant with the release of the DNA and transcription initiation (Fig. 1d). It has been shown that the *Thermus thermophilus* (*Tt*) CBD domain has exciting biotechnological potential, including the formation of light-responsive hydrogels for drug delivery[14–17], smart

wearable technology devices[18] and regulation of the expression of mammalian genes[19]. The isolated *Tt*CarH CBD was reported to form a stable tetramer and undergo light-driven monomerization in the above-mentioned applications.

Crystal structures of the dark and light-adapted state of *Tt*CarH have been reported[12]. In the dark state, a conserved His177 residue from the CBD Rossman fold domain directly coordinates the cobalt atom of the corrin ring in a base-off/His-on conformation (Fig. 1e). The B$_{12}$-adenosyl moiety forms interactions with the conserved Trp131 (stacking) and Glu141 (hydrogen bond) from the CBD four-helix bundle region. Residues from the DBD do not participate in AdoCbl binding. Following light activation, the AdoCbl Co-C bond photolyses and the adenosyl moiety leaves the pocket as 4′,5′-anhydroadenosine[20]. This triggers a series of events that ultimately lead to the formation of a bis-His ligated cobalamin with the upper axial ligand contributed by

[1]Manchester Institute of Biotechnology, Department of Chemistry, University of Manchester, Manchester, UK. [2]Univ. Grenoble Alpes, CEA, CNRS, Institut de Biologie Structurale, F-38044 Grenoble, France. [3]Photon Science Institute, Department of Chemistry, University of Manchester, Manchester, UK. ✉e-mail: nigel.scrutton@manchester.ac.uk; david.leys@manchester.ac.uk

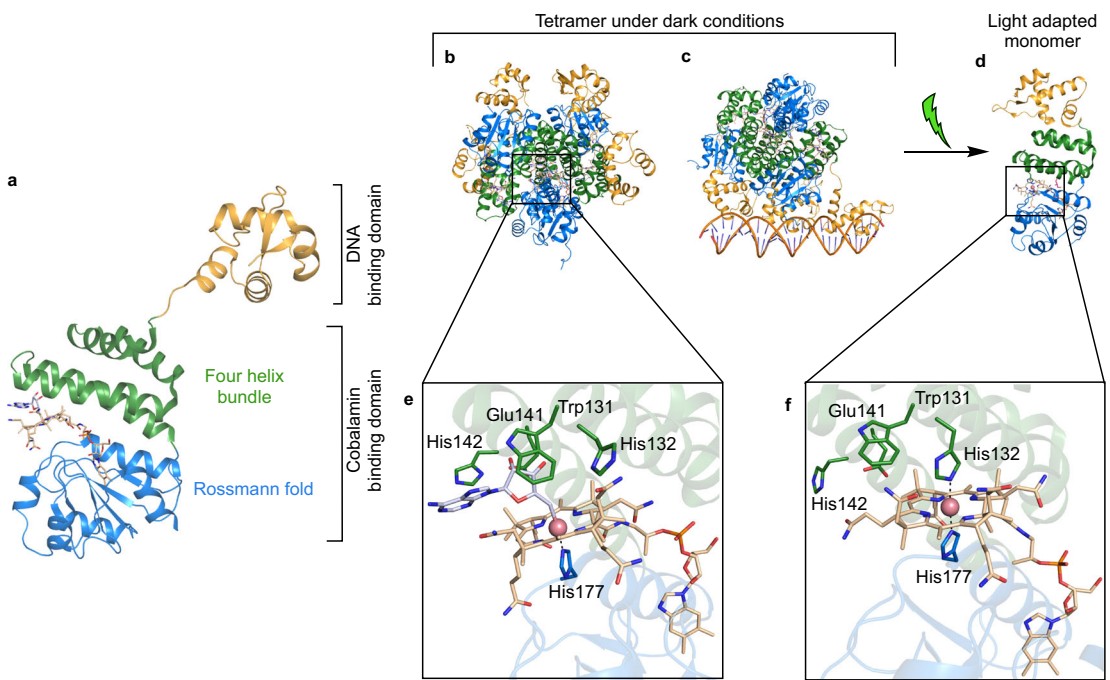

**Fig. 1 | Crystal structure of CarH in dark- and light-adapted states. a** Cartoon representation of a full-length dark monomer, showing the different domains and the binding location of AdoCbl (shown as sticks) at the interface of subdomains of CBD (PDB 5C8D). **b, c** Dark tetramer capable of binding DNA via the DBD (shown in orange for all monomers) (PDB 5C8E). CBD region coloured in green and blue. **d** Light-adapted monomer obtained following photolysis of the Co-C bond in AdoCbl and tetramer dissociation in solution (PDB 5C8F). **e, f** Close-up view of the AdoCbl binding site in individual monomers of dark-state (**e**) and light-adapted (**f**) CarH[12]. The residues participating in important interactions are shown as sticks. Cobalamin is shown in beige and the adenosyl moiety is depicted in grey.

His132 as shown in a crystal structure of the light-adapted *Tt*CarH monomer (Fig. 1f)[12]. Movement of His132 to form bis-His ligation is thought to reorient the four-helix bundle closer to the Rossmann fold and trigger disruption of the tetramer interface[12]. However, the exact nature and sequence of events remains unclear, as structural description of intermediate species is currently lacking.

Time-resolved absorption spectroscopy has provided important insights into the formation of different intermediates during the CarH photocycle at timescales ranging from ps to s[21,22]. Kutta et al. suggested that the Co-C bond scission is likely to proceed via a heterolytic cleavage route, leading to further distinct spectral intermediates on the μs-ms timescales and tetramer dissociation over several seconds[21]. It has also been proposed a long-lived triplet state is formed upon excitation, which ultimately reacts to form the stable 4′, 5′ anhydroadenosine product and final light state of CarH[22]. Furthermore, it has been shown that oxygen is required for full conversion to the light-adapted state, while under anaerobic conditions CarH forms a stable Co(II) intermediate species upon light activation[20]. Outstanding questions regarding the photocycle remain, with both the structural dynamics associated with the transition of dark-adapted tetramer to light-adapted monomer and the role of oxygen poorly understood.

Here, we report crystal structures of the *Tt*CarH CBD (referred to as *Tt*CBD) and a H132A variant following illumination under both aerobic and anaerobic conditions, but prior to tetramer dissociation. The crystal structures obtained, combined with solution absorbance and EPR spectroscopy studies, reveal that the conformational changes are driven by unexpected Co-protein ligation changes in response to distinct Co redox states. This involves disruption of the lower His177-Co coordination in favour of His132-Co ligated structures, as part of modest conformational changes that initially occur upon illumination in crystallo. Computational studies suggest the ligand switch is associated with the weakening of the oligomeric interaction as a prelude to tetramer dissociation. While Co oxidation ultimately drives monomer formation and bis-His ligation in the WT in solution, tetramer dissociation is also dependent on the presence of the upper His ligand and does not occur in an H132A variant. These results provide important insights into the light-driven structural changes that underpin the transition to the light-adapted state of B₁₂ photoreceptors and reveal CarH control of carotenoid biosynthesis integrates sequential response to light and oxygen.

## Results

### Role of Co(I) and Co(II) redox states upon aerobic and anaerobic photolysis

The AdoCbl-bound *Tt*CBD showed similar light-activated spectroscopic behaviour to the full-length *Tt*CarH protein. The *Tt*CBD formed a stable tetramer in the dark (Supplementary Figs. 1 and 2a), with a similar UV-vis absorbance spectrum as previously reported[21] (Fig. 2a and Supplementary Fig. 3b). Upon illumination, the *Tt*CBD forms a similar monomeric light-adapted state (Supplementary Figs. 1 and 2b) as the full-length protein (Supplementary Figs. 2f and 13a), with a characteristic bis-His ligated absorbance peak at 359 nm (Fig. 2a).

It has previously been shown that under anaerobic conditions, full-length *Tt*CarH forms a Co(II) cobalamin upon extended exposure to ambient light[20]. We have explored the anaerobic photoconversion in more detail in *Tt*CBD, which initially yielded an absorbance spectrum with an intense absorbance band at 388 nm and weak absorbance peaks at 455, 545 and 680 nm of the visible spectral region (Fig. 2b). These spectral features, together with the almost colourless nature (Fig. 2d) of the light-exposed anaerobic sample, are indicative of a Co(I) cobalamin species[23,24]. The lifetime of the Co(I) cobalamin species is sensitive to the buffer system used and increases in phosphate buffer, likely due to more inert nature of phosphate buffer system vis-a-vis Tris buffer (Supplementary Fig. 3a). A similar transient Co(I) species was also observed with the full-length CarH protein (Supplementary Fig. 3b). Within ~1 h, the highly reactive Co(I) supernucleophile converts entirely to a stable orange-coloured species (Figs. 2b, d), with absorbance features at 475 and 310 nm that are

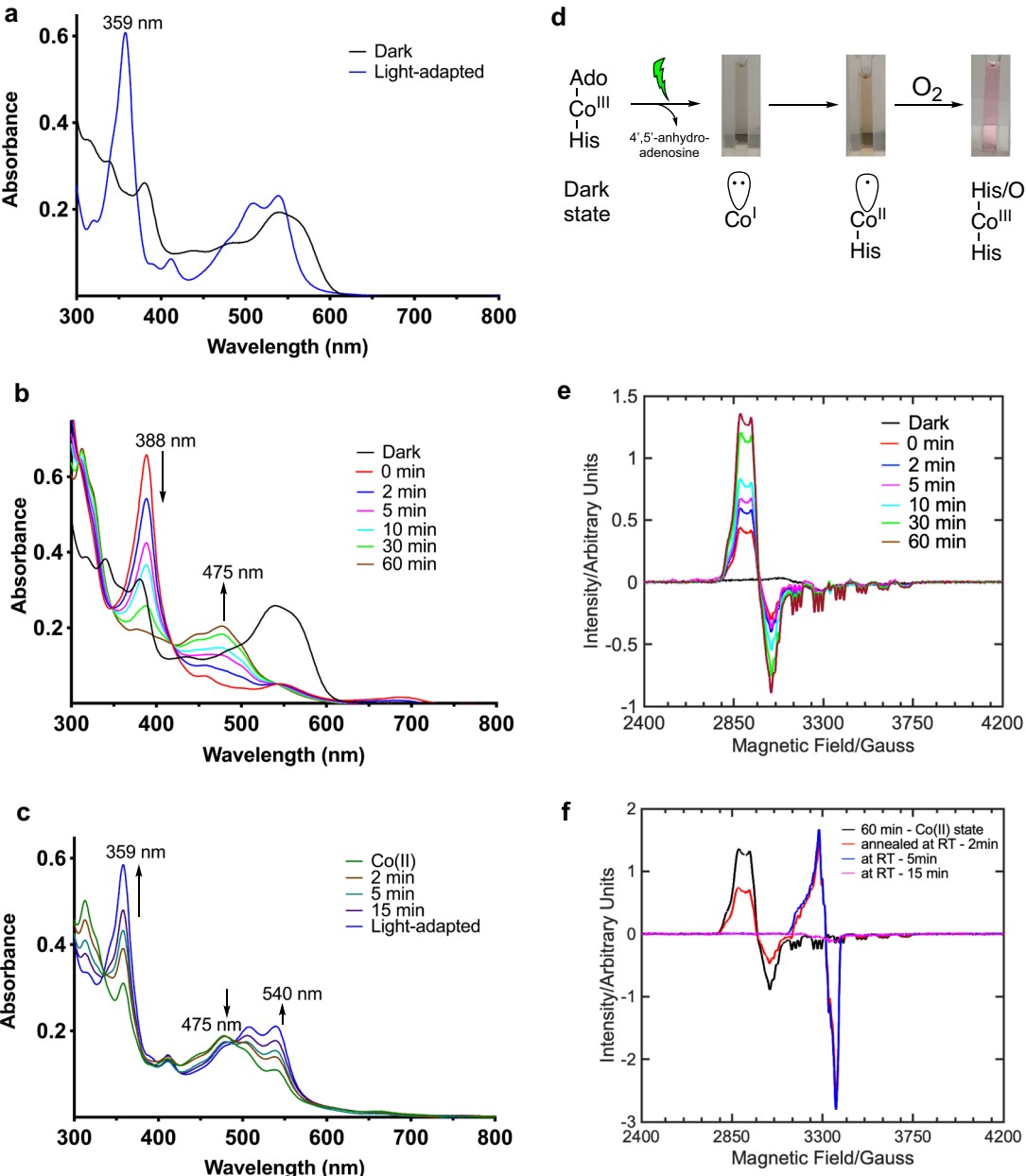

**Fig. 2 | UV-Vis and EPR spectroscopy for wild-type *Tt*CBD.** Photoconversion followed by UV-Vis spectroscopy under **a** aerobic and **b** anaerobic conditions in 50 mM Tris-HCl buffer (pH 7.5). The times indicated in the key refer to the length of time samples were incubated in the dark following illumination for 15 s. **c** Gradual conversion of Co(II) species to light-adapted state on exposure to O₂ at room temperature. **d** Observed change in colour of the sample due to the formation of different Co oxidation states in solution during the photocycle. The

hexacoordinate Co(III) state can have a 6th coordination as His or OHₓ ligand ($x = 1$ or 2). **e** The light-exposed anaerobic *Tt*CBD samples were flash-frozen in liquid nitrogen following incubation in the dark at room temperature for various periods of time, with the corresponding cw-EPR spectra measured at 20 K confirming the formation of Co(II) species. **f** Annealing of the light-exposed, anaerobic *Tt*CBD sample at RT shows a conversion of Co(II) to Co(III)-superoxo and subsequently to Co(III) light-adapted state as followed by EPR spectroscopy.

characteristic of Co(II) cobalamin[20,24,25]. This stable Co(II) cobalamin species is readily converted to the light-adapted Co(III) state upon exposure to O₂[26], with formation of the characteristic pink-coloured bis-His ligated species, with an absorbance peak at 359 nm (Fig. 2c, 2d, Fig. 3d).

The initial formation of the Co(I) from the Ado-Co(III)-His177 dark state through anaerobic illumination and subsequent conversion to the Co(II) cobalamin species were confirmed using EPR spectroscopy (Supplementary Fig. 4). Anaerobic samples that were photolysed immediately prior to flash-cooling contained only small amounts of the paramagnetic Co(II) species. However, there was a gradual increase in the Co(II) signal over time, with full conversion to Co(II) cobalamin

species in ~60 min at room temperature in the dark (Fig. 2e), in good agreement with the UV-vis absorbance data (Fig. 2b). The EPR spectrum showed hyperfine and superhyperfine splitting which can be attributed to the ⁵⁹Co and axially bound ¹⁴N-nuclei of the histidine ligand respectively[20] (Supplementary Fig. 5). Exposure of the *Tt*CBD sample at room temperature to air leads to a slow conversion over several minutes into the final light-adapted, Co(III) bis-His ligated cobalamin species via the formation of a Co(III)-superoxo species[27,28] (Fig. 2f and Supplementary Fig. 5).

As the Co(I) and Co (II) states have not previously been observed in time-resolved spectroscopy measurements under fully aerobic conditions[21] we sought to explore their potential roles in the aerobic

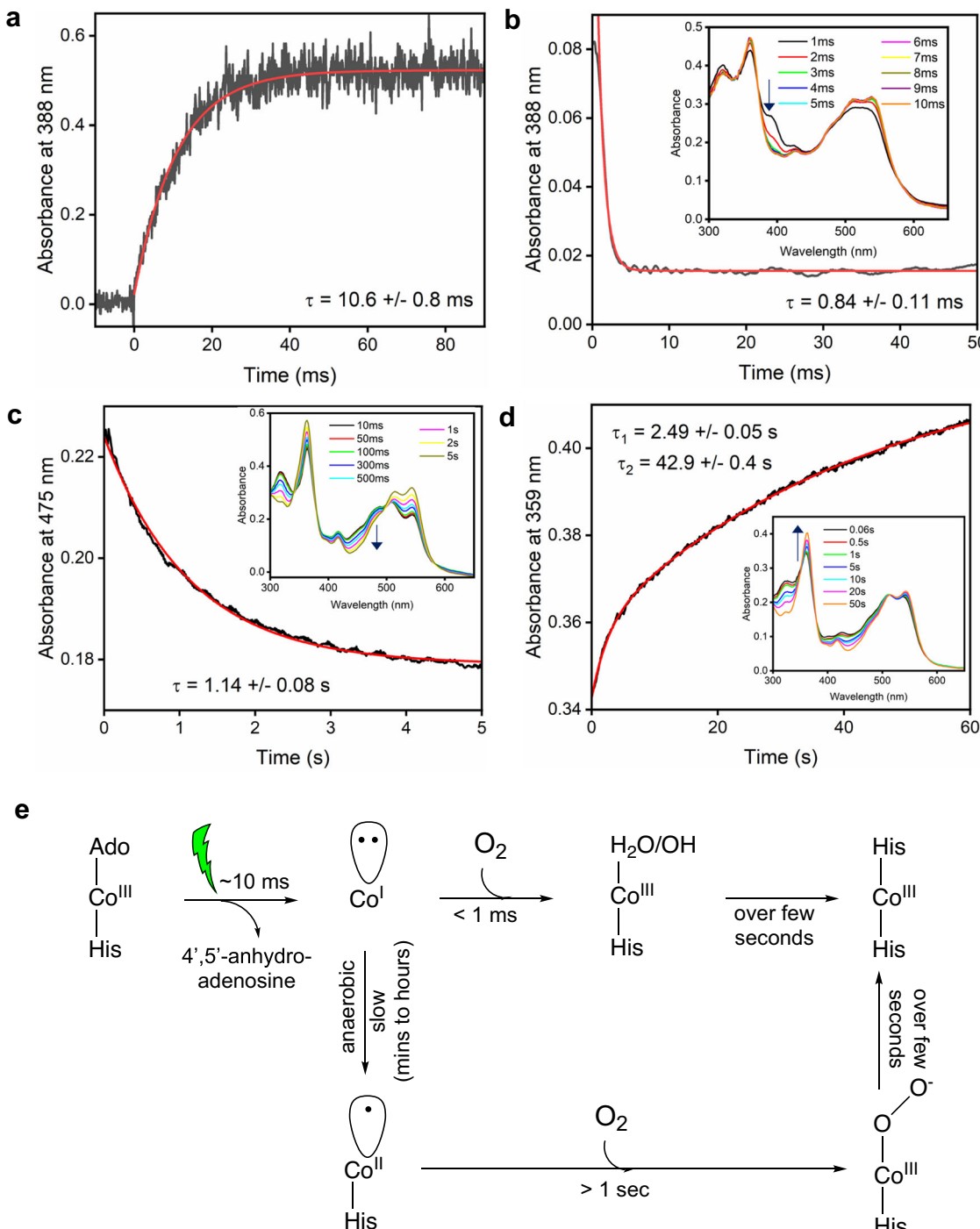

**Fig. 3 | Kinetics of the formation and decay of Co(I) and Co(II) states using stopped flow. a** Kinetic transient at 388 nm showing rate of formation of Co(I) for anaerobic *Tt*CBD sample after photoexcitation with a laser pulse at 530 nm (15 mJ). **b** Kinetic transient at 388 nm showing rate of decay of Co(I) species upon rapid mixing of anaerobically photolysed *Tt*CBD sample with oxygenated buffer from stopped-flow experiments. The inset shows the corresponding absorbance spectra over a range of timepoints. **c** Kinetic transient at 475 nm showing rate of decay of Co(II) species upon rapid mixing of anaerobically photolysed *Tt*CBD sample with oxygenated buffer from stopped-flow experiments. The inset shows the corresponding absorbance spectra over a range of timepoints. **d** Kinetic transient at 359 nm showing formation of final bis-His ligated light state over several seconds upon rapid mixing of anaerobically photolysed *Tt*CBD sample with oxygenated buffer from stopped-flow experiments. The inset shows the corresponding absorbance spectra over a range of timepoints. **e** Scheme describing the changes in Co redox state following light activation under distinct [O₂] conditions.

photolysis mechanism (Fig. 3). Laser photoexcitation measurements under anaerobic conditions show that the Co(I) intermediate forms with a lifetime of ~10 ms (Fig. 3a). Conversely, exposure of a Co(I) sample to O₂ results in substantially faster formation of a Co(III) intermediate (<1 ms lifetime, Fig. 3b), thus highlighting why the Co(I)

state is not observed under fully aerobic conditions[21]. In contrast, the similar reaction of the Co(II) state with O₂, presumably to form the Co(III) superoxo species, is much slower with a lifetime of >1 s (Fig. 3c). Hence, although both Co(I) and Co(II) species can react productively with O₂, the preferred route under fully aerobic conditions would be

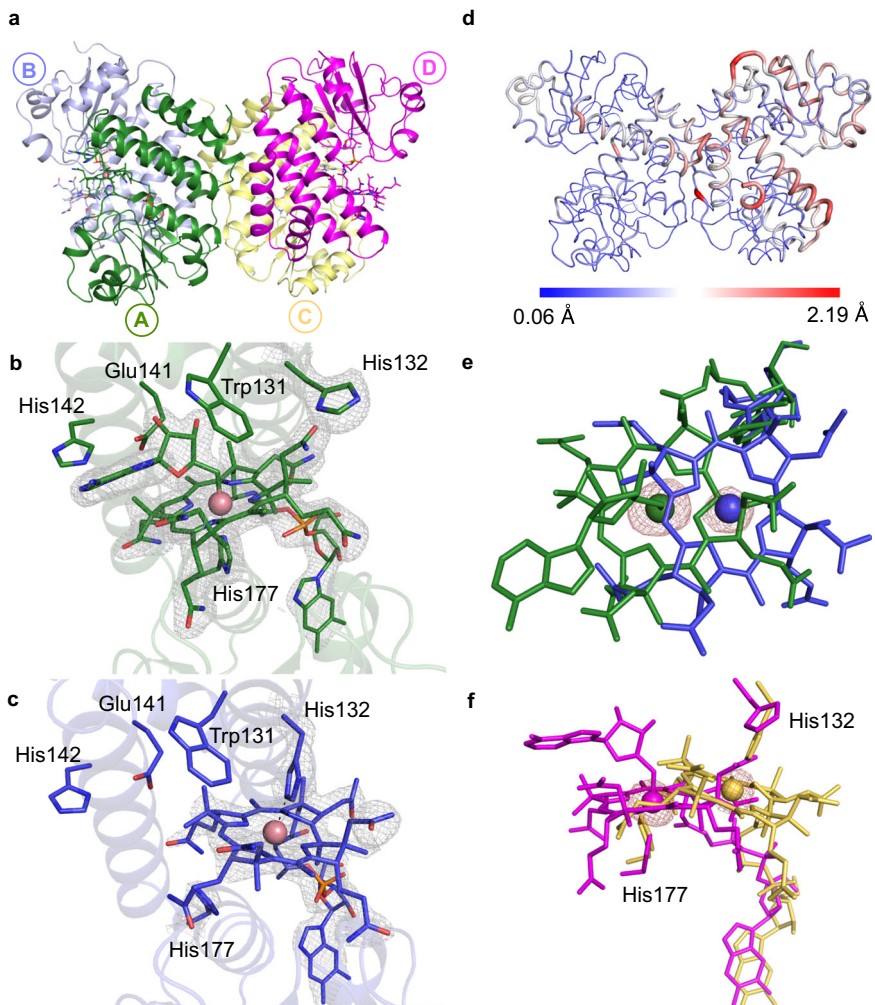

**Fig. 4 | Anaerobic photoconversion in crystallo of wild type *Tt*CBD in form 1 crystals. a** Cartoon representation of form 1 D^O2 wild type *Tt*CBD tetramer structure. Individual monomers are shown in different colours and the bound AdoCbl is shown as sticks. The chain nomenclature is shown in circles. Chains A and C are involved in dimer-dimer interaction and crucial for tetramer assembly. **b, c** A side by side comparison of the D^O2 (**b**) and the I^anaer (**c**) structures obtained from form 1 crystals. The grey mesh depicts the composite omit 2F_o–F_c electron density

contoured at 1.5 σ for bound AdoCbl/cobalamin and His 177/132. The Co-ligand interaction is shown as dashed lines. **d** The displacement for Cα atoms in the I^anaer structure with respect to the corresponding D^anaer is shown. **e, f** Comparison of cobalamin binding in **e** chain A and **f** chain D of D^O2 (green and purple) and I^anaer structures (blue and yellow) showing the extent of corrin ring displacement. The red mesh shows the anomalous density for the Co atom contoured at 3.0 σ.

via the Co(I) state. The final bis-His ligated light state is only formed slowly over several seconds (Fig. 3d), in agreement with previous work by Kutta et al., who also showed that the final bis-His ligated state was preceded by a species with an absorbance maximum at 352 nm under aerobic conditions[21]. This suggests that final bis-His ligation, which is presumably coupled to monomer formation, is achieved following Co(I) oxidation via distinct intermediates, with separate routes leading to either transient Co(II) or Co(III) species (Fig. 3e). However, it should be noted that alternative mechanisms where photoexcitation in the presence of O_2 does not result in the formation of a Co(I) state cannot be ruled out.

**Structural changes accompany Co(II) formation under anaerobic conditions**

We solved structures of the *Tt*CBD dark state under aerobic conditions (Dark aerobic referred to as D^O2) for two distinct crystal forms, to 1.8 Å (form 1, pH 6.5; Supplementary Table 1; PDB 8C31) and 1.7 Å (form 2, pH 7.5; Supplementary Table 2; PDB 8C73) resolution, respectively. In both cases, a complete tetramer is present in the asymmetric unit (Fig. 4a depicts the form 1 D^O2 structure). Both D^O2 structures

superpose with the full-length *Tt*CarH structure[12] (PDB−5C8D) with an average r.m.s.d. of ~0.44 Å and ~0.54 Å for all Cα atoms of form 1 (Supplementary Fig. 6a) and 2 respectively, confirming that the DBD truncation does not affect tetramer assembly. The crystal packing and symmetry contacts for both crystal forms are nearly identical, with a r.m.s.d of ~0.29 Å (for all Cα backbone atoms (Supplementary Fig. 7)). The data shows clear electron density for AdoCbl in the four cobalamin binding sites for both crystal forms (Fig. 4b depicts the form 1 D^O2 structure, Supplementary Fig. 6b; form 2 D^O2 structure shown in Supplementary Fig. 8a, d, g, j). The His132 residue, which contributes to the upper axial ligand in the light-adapted full-length structure[12] is pointing away from the binding pocket (Fig. 4b).

Guided by our earlier spectroscopic studies, we aimed to provide a structural basis for the formation of intermediate cobalamin redox states in the absence of O_2, by preparing form 1 crystals of wild-type *Tt*CBD under anaerobic conditions. The structure obtained from the dark-state anaerobic crystals (2.2 Å resolution, PDB 8C32, referred to as D^anaer) is highly similar to the D^O2 crystal structure (Supplementary Fig. 6c). When anaerobic crystals were exposed to light, a visible change in the colour of the crystals from pink to orange was observed

without loss of diffraction power (Supplementary Fig. 9). Intriguingly, some unit cell parameters are perturbed by ~3.0 Å in response to illumination (Supplementary Table 1), suggesting a minor lattice rearrangement occurred in crystallo. The corresponding 2.2 Å illuminated structure (PDB 8C33, referred to as I$^{anaer}$) reveals a lack of electron density for the adenosyl moiety in the four cobalamin binding sites. Unexpectedly, the lower axial His177-Co bond is broken, concomitant with corrin ring reorientation (Fig. 4c). As a consequence, the penta-coordinate Co atom is ligated by His132, which has moved inwards to form the new His132-Co bond (Fig. 4c). Anomalous maps further confirm the change in position of the Co atom (Fig. 4e, Supplementary Fig. 10b) by ~3.4 Å for monomer A, ~3.7 Å for monomer B, ~5.1 Å for monomer C and ~4.5 Å in case of monomer D. In addition, the B$_{12}$ dimethylbenzimidazole (DMB) tail also repositions while retaining hydrogen bonding interactions observed in the dark structure (Supplementary Fig. 10c). One of the amide groups adorning the corrin ring makes new hydrogen bonding interactions with Ser226 and Glu227, while another amide is within hydrogen bonding distance of Gln129 and Gln249 (Supplementary Fig. 11a). The displaced corrin ring in the I$^{anaer}$ structure thus relinquishes the interactions with the Rossmann fold domain, and establishes interactions with the His132-containing four-helix bundle domain that are similar to those observed in light-adapted monomeric CarH structure (PDB 5C8F)[12] (Supplementary Fig. 11b). Superposition of the individual domains from both light-adapted structures (i.e. anaerobic tetramer and aerobic monomer) suggests significant relative movement of both domains is required to attain the final bis-His ligated state (Supplementary Fig. 11c, d).

A detailed comparison of D$^{O2/anaer}$ and I$^{anaer}$ tetrameric structures reveals modest and distinct structural adaptation for the four monomers in response to photolysis of Co-C bond and reorientation of corrin ring. The average displacement for Cα atoms in monomer D is ~1.0 Å, which is considerably higher than the other chains (Fig. 4d, Supplementary Fig. 10a, Supplementary Table 3). This is likely due to the relative lack of crystal contacts for monomer D, allowing a higher degree of flexibility (Supplementary Fig. 7b) and reflected in larger cobalamin displacement of ~5.1 Å (Fig. 4f, Supplementary Fig. 10d, e). The cobalamin movement requires the long helix (residues 113–134) in the four-helix bundle to move away from the position adopted under dark conditions (Fig. 4d). The Cα atoms in this helix, which also contains the conserved Trp131 residue, show a higher average displacement of ~1.4 Å (Supplementary Table 3). The loop region (residues 157–167) linking the four-helix bundle and Rossmann fold, and a small loop (residues 172–179) containing the conserved His177 residue also show a higher than average displacement of 1.4 Å and 1.1 Å, respectively. The effect of these local changes on the overall assembly is minimal, and there is no significant change observed in the radius of gyration computed from the overall tetramer model in the I$^{anaer}$ model when compared to the D$^{O2/anaer}$ models.

## Cobalamin ligand exchange affects the oligomeric interface

The CarH tetramer contains two distinct monomer interfaces (Fig. 4a). At the A/B and C/D dimer interface, residues Arg176 and Asp201 form a polar network crucial for dimer integrity[12]. Intriguingly, in the I$^{anaer}$ structure, the Arg176-Asp201 interaction switches from inter-monomeric to intra-monomer (Fig. 5a). The Asp201 side chain makes a water-mediated hydrogen bond with the N1 adenosyl moiety in the dark structure (Fig. 5a). This interaction is lost in the I$^{anaer}$ structure as a consequence of the adenosyl group leaving the binding pocket. These perturbations are observed for both A/B and C/D dimers.

The dimer-dimer interface formed between monomers A and C also shows some subtle changes (Fig. 5b). Most notably, in monomer C only, the conserved Gly160 that is located on the loop connecting the four-helix bundle and Rossmann fold, and reported to be important for dimer-dimer contact[12], moves away from the A-C interface (Fig. 5b inset). This results in a ~2 Å distance increase between the A/C Gly160

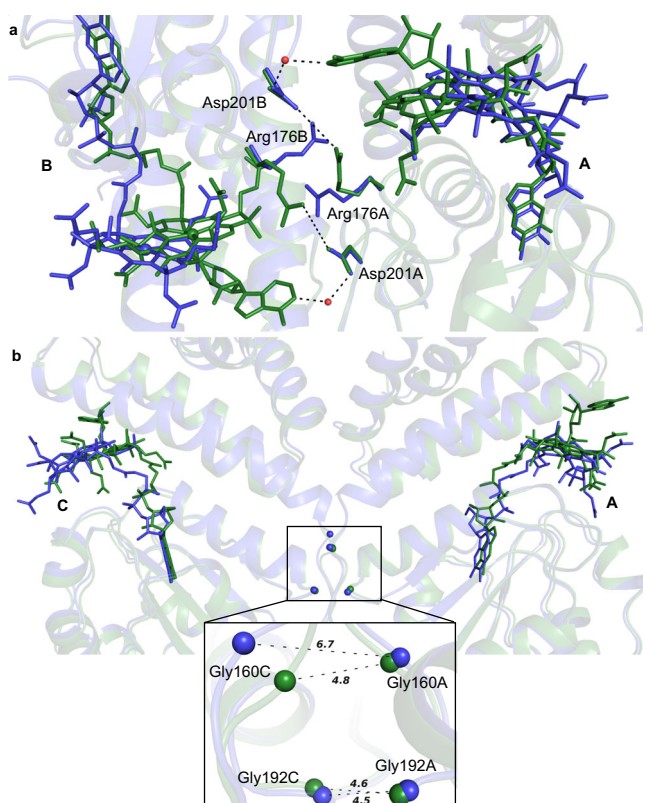

**Fig. 5 | Comparison of wild-type D$^{O2}$ (green) and I$^{anaer}$ (blue) structures obtained from form 1 crystals. a** Interface of monomers A and B involved in dimer formation and **b** monomers A and C involved in dimer-dimer contacts. The backbone for individual chains is shown as cartoon. Water molecules are shown as red spheres. Inset shows close-up view of the dimer-dimer interface formed by chain A and C. The distance between Gly160 and Gly192 Cα atoms is shown as dashed lines. Letters in bold are chain identifiers.

Cα atoms in the I$^{anaer}$ structure. It is likely that the loop harbouring Gly160 acts as a hinge point on which the two respective domains reorients themselves to achieve the final bis-His ligated state.

The electrostatic potential maps of the two dimers show strong complementarity at the dimer interface, which suggests that any structural changes that perturb the electrostatics can have a significant effect on the binding energy. To estimate how the observed movement of individual chains and certain residues in the wild-type I$^{anaer}$ model ultimately translates into monomer formation, we computed the electrostatic forces and binding free energies (using DelphiForce[29]) for the corresponding dimer and tetramer assemblies. Electrostatic forces between the monomers in each dimer reduced by ~25%, whereas those between the two dimers in the tetramer decreased by as much as 66% (Supplementary Table 4). The binding energy also saw a decrease of 36% for the overall tetramer assembly (Supplementary Fig. 12).

## His-ligation switch also occurs in crystalline TtCBD illuminated under aerobic conditions

Unfortunately, in the presence of oxygen, prolonged illumination (>5 s) of either WT TtCBD crystal form under ambient relative humidity conditions is associated with dramatic loss of diffraction resolution, presumably due to disruption of the lattice order as a possible consequence of tetramer dissociation. Hence, in order to establish whether the light-triggered His-ligation switch occurs under aerobic conditions, we exposed both TtCBD crystal forms to 530 nm wavelength light (LED) for periods ranging from ~1 s to 5 s, followed by rapidly flash cooling in liquid nitrogen (structures referred to as I$^{O2}$). Form 1 I$^{O2}$ crystals diffracted up to 1.8 Å (PDB 8C34) and revealed a

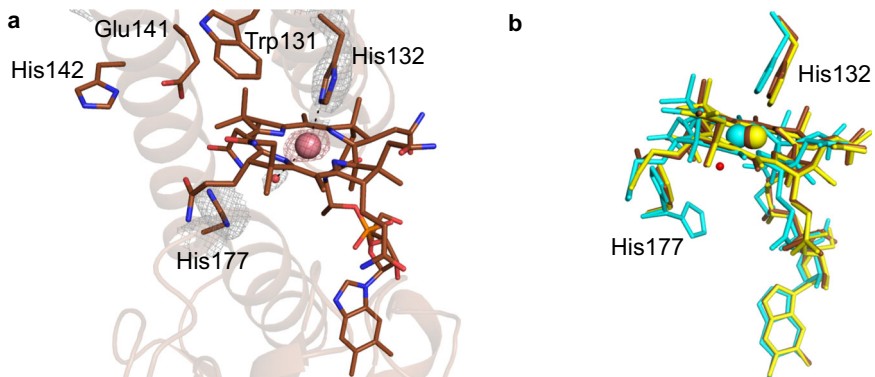

**Fig. 6 | Photoconversion of *Tt*CBD in form 1 and 2 crystals in crystallo under aerobic conditions. a** Stick representation for the most displaced cobalamin (i.e. chain D) of I$^{O2}$ for crystal form 1. The grey mesh depicts the composite omit 2F$_o$–F$_c$ density contoured at 1 σ, and the interactions made with the Co atom shown as dashed lines. The red mesh shows the anomalous density for the Co atom contoured at 3.0 σ. **b** Comparison of the displaced cobalamin of illuminated structures for form 1 (brown I$^{O2}$ and yellow I$^{anaer}$; chain D) and form 2 (cyan I$^{O2}$; chain A).

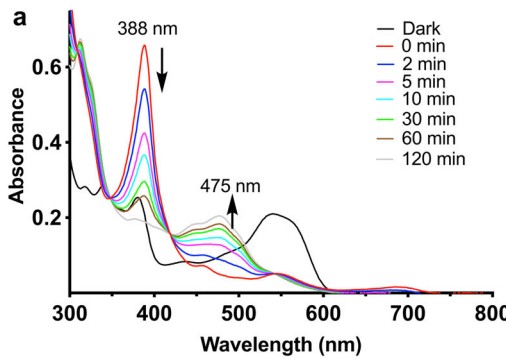

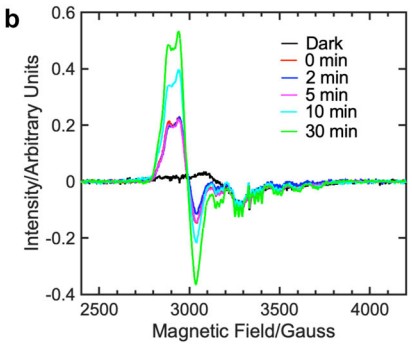

**Fig. 7 | UV-Vis and EPR spectroscopy for the *Tt*CBD-H132A variant.**
**a** Photoconversion under anaerobic conditions in 50 mM Tris-HCl buffer (pH 7.5). The times indicated in the key refer to the length of time samples were incubated in the dark following illumination for 15 s. **b** The light-exposed, anaerobic *Tt*CBD-H132A samples were frozen in liquid nitrogen as a function time and the corresponding cw-EPR spectra, measured at 20 K.

similar unit cell perturbation as observed for the I$^{anaer}$ crystals. In the case of form 2 I$^{O2}$ crystals, a 2.5 Å resolution (PDB 8C76) structure was obtained (Supplementary Table 2), which aligns with a r.m.s.d of ~0.43 Å to the form 1 I$^{O2}$ structure. The form 1 I$^{O2}$ structures overlay with a r.m.s.d. of only 0.17 Å compared to the corresponding I$^{anaer}$ structure, and inspection of the electron density map confirmed a similar Co-His177 to His132-Co-ligation switch has occurred. Chains A, B and D show clear density for the repositioned cobalamin moiety (Fig. 6a) and associated His132 coordination (Fig. 6b). In the form 2 I$^{O2}$ structure, there is a similar shift in the cobalamin and associated His132 position in chains A, B and C, but there is no discernible density for His132 in chain D (Supplementary Fig. 8b, e, h, k). In the higher resolution I$^{O2}$ structure (form 1), and unlike the I$^{anaer}$ structure, density consistent with a water molecule can be observed at ~2.8 Å from the Co in three of the four monomers comprising the tetramer. The average His132-Co distance increases by ~0.45 Å when comparing the I$^{anaer}$ (average distance 2.5 Å) with the corresponding I$^{O2}$ structure (average distance 2.95 Å). The increase in His132-Co bond length likely correlates with a change from pentacoordinate to hexacoordinate state in the presence of oxygen. Both I$^{O2}$ models show a similar movement of the chain C Gly160 residue at the tetramer A/C interface.

## The role of His132 in photoactivation

The role of His132 in light-driven structural rearrangement was studied by producing a TtCBD-H132A variant. Although a H132A variant of the full-length *Tt*CarH[12] has previously been reported to undergo tetramer disassembly upon illumination under aerobic ambient conditions, we find that the majority of the aerobic light-exposed *Tt*CarH H132A (both full-length and CBD-only) remain in the tetrameric state (Supplementary Fig. 1, Supplementary Fig. 2d and h, Supplementary Fig. 13b and d) in comparison to the WT protein under identical illumination conditions. The absorbance spectrum for the dark state of *Tt*CBD-H132A resembles that of wild-type *Tt*CBD, whereas the light-adapted spectrum shows an absorbance peak at 352 nm instead of 359 nm, indicating formation of a hydroxylated Co(III) species instead of the bis-His ligated state also reported previously for the full-length H132A variant[12] (Supplementary Fig. 13d, Fig. 14a, b).

Anaerobic photoconversion of the Ado-Co(III)-His177 ligated *Tt*CBD-H132A shows a spectrum which is consistent with the formation of a Co(I) cobalamin species (Fig. 7a). The rate of decay of this species is significantly slower than wild type *Tt*CBD, with complete conversion to Co(II) species observed in ~120 min under anaerobic conditions at room temperature in Tris-HCl buffer (Fig. 7a). The rate of decay of Co(I) cobalamin species is significantly slower in phosphate buffer when compared to Tris buffer conditons (Supplementary Fig. 14c). The characteristic Co(II) EPR signals ($^{14}$N-superhyperfine structure along the $g_z$ position) of the *Tt*CBD-H132A suggest that Co(II) is in a five-coordinate geometry with a nitrogen ligand in one of the axial positions (Fig. 7b). As observed in wild type *Tt*CBD, annealing of the *Tt*CBD-H132A at room temperature in air shows conversion of the Co(II) via Co(III)-superoxo to the light-adapted Co(III) oxidation state (Supplementary Fig. 14d and 15) within 20 min.

The 2.1 Å *Tt*CBD-H132A dark state structure (PDB 8C35) obtained under aerobic conditions (referred to as D$^{O2-H132A}$) reveals both the

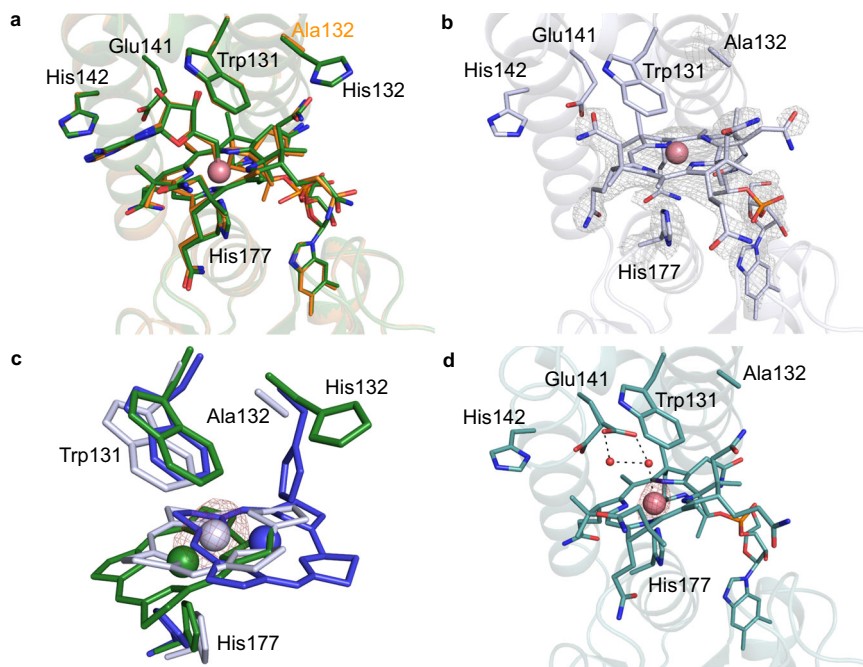

**Fig. 8 | Form 1 crystal structures of the *Tt*CBD-H132A variant. a** Superposition of wild type $D^{O2}$ (PDB 8C31 green) and $D^{O2\text{-}H132A}$ (PDB 8C35 orange) models showing residues and AdoCbl binding. **b** Electron density of bound tetracoordinate cobalamin in $I^{anaer\text{-}H132A}$ structure. The grey mesh depicts the composite omit $2F_o\text{-}F_c$ density contoured at 1.5 σ. **c** Comparison of corrin ring position in wild-type $D^{O2}$ (green), $I^{anaer}$ (blue) and $I^{anaer\text{-}H132A}$ (grey) structures. The red mesh shows the anomalous density for Co atom contoured at 3.0 σ. **d** The $I^{O2\text{-}H132A}$ structure shows a water ligated hexa-coordinated His177-Co(III)-$H_2O$.

tetramer conformation and AdoCbl binding mode (Fig. 8a and Supplementary Fig. 16a) are unaffected by the H132A mutation (Supplementary Fig. 16d). Photoconversion of anaerobic *Tt*CBD-H132A crystals yielded a distinct colour change to the WT and the corresponding 2.15 Å structure (PDB 8C37) reveals the formation of a tetra-coordinated corrin species (referred to as $I^{anaer\text{-}H132A}$). Here, the His177-Co bond has been broken (~3.8 Å distance for chain A) with an outward displacement of the corrin ring, similar to that observed in wild-type $I^{anaer}$ structure (Fig. 8b). Anomalous density confirmed the location of the Co atom in chain A (Fig. 8c). This structure is likely to represent the Co(I) cobalamin state, which is more stable in the *Tt*CBD-H132A variant (Fig. 7a and Supplementary Fig. 14c). Two distinct but closely spaced Co anomalous signals were visible in the other monomers, suggesting multiple binding modes for cobalamin (Supplementary Fig. 17).

Compared to the wild type, the superposition of the $I^{anaer\text{-}H132A}$ structure with the corresponding $D^{O2\text{-}H132A}$ showed much less global structural perturbation in the overall protein backbone (average displacement Cα atoms – 0.4 Å, Supplementary Figs. 18a and 16e and Supplementary Table 3). Furthermore, in contrast to the wild type, aerobic *Tt*CBD-H132A crystals can withstand light exposure for prolonged periods. In this case, the unit cell parameters are not perturbed by illumination (Supplementary Table 1) and the corresponding 2.0 Å structure (PDB 8C36, referred to as $I^{O2\text{-}H132A}$) resembles the wild type $D^{O2}$ structure (Supplementary Fig. 16f) with a His177 ligated cobalamin but lacking any discernible electron density for the adenosyl moiety (Fig. 8d, Supplementary Fig. 16b). The residual density above the corrin ring is consistent with that of a water molecule, indicating the presence of a hexa-coordinated $OH_x$ ligated Co(III) species. In all four chains, the conserved Glu141 from the EH motif adopts two different conformations (Fig. 8d), while other residues lining the AdoCbl binding pocket are unperturbed (Supplementary Fig. 16c). The average Cα displacement compared to wild-type dark structure is only ~0.35 Å (Supplementary Table 3, Supplementary Fig. 18b), with only the conserved Gly160 from the loop region connecting the four-helix bundle and Rossmann fold domains in monomers B and D displaced by ~2.0 Å.

While this residue makes key contributions to the chain A/C interface[12], neither Gly160 belonging to chains A or C displays the altered conformation observed in B and D.

## Discussion

An in-depth understanding of the structural transitions that underpin the photoresponse in $B_{12}$-dependent photoreceptors has been lacking. Crystal structures of the dark and light states of the canonical CarH protein reveal formation of a bis-His ligated state is associated with tetramer dissociation into the photoactivated monomeric form of the protein[12]. By combining structural biology on two crystal forms with solution spectroscopy and computation, the present work indicates that a series of redox-driven Co-ligation changes occur following photolysis and displacement of the upper axial adenosyl ligand (Fig. 9, Supplementary Movie 1). Illumination under anaerobic conditions leads to conversion of the lower axial His177-Co(III)-Ado ligated species to pentacoordinate Co(II)-His132 upper axial ligation (observed in WT $I^{anaer}$), presumably via a tetracoordinate Co(I) species (as seen in $I^{anaer\text{-}H132A}$). Our spectroscopic studies confirm $O_2$ is required to form the final light-activated hexa-coordinate His177-Co(III)-His132 state of CarH, which is hypothesised to be accompanied by domain movement of the Rossmann fold and four-helix bundle to achieve the bis-His ligation in the wild type protein. Illumination of both WT *Tt*CBD crystal forms under aerobic conditions combined with rapid flash-cooling reveals a similar ligand-switch occurs, suggesting fast light-triggered photochemistry gates the lower redox-driven Co-protein ligation changes, with a tetrameric $H_2O$-Co(III)-His132 likely preceding monomeric His177-Co(III)-His132 formation (Fig. 9a). In the *Tt*CBD-H132A variant, light-induced protein structural changes are minimal and His177-Co bond breakage is not observed, suggesting the CarH conformational changes are directly coupled to the cobalamin redox state through His132 ligation (Fig. 9b).

The new His132 ligated cobalamin binding mode observed in the wild-type $I^{O2/anaer}$ structures resembles that proposed from previous molecular dynamics simulations[21]. It provides further evidence that the

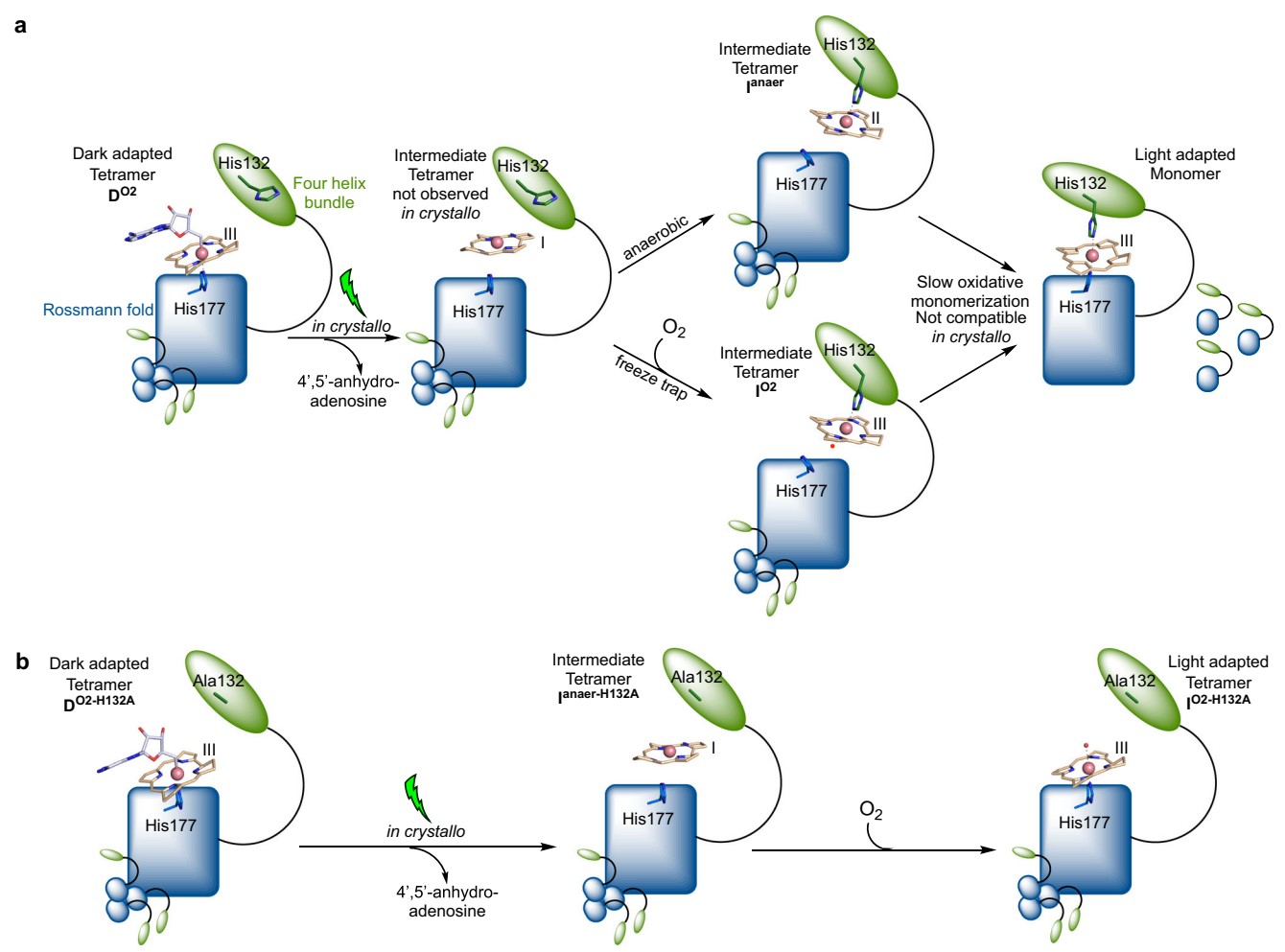

**Fig. 9 | Cartoon representation of cobalamin conformational changes following CarH illumination.** Changes shown for **a** wild type *Tt*CBD and **b** H132A variant. The redox state of the Co atom is shown in roman numerals. Only the corrin ring structure belonging to each state is shown for ease of visualisation. Supplementary Movie 1 illustrates the proposed conformational changes.

adenosyl moiety in AdoCbl plays a critical role in guiding the corrin ring towards His177 via stacking interactions with the conserved Trp131 residue. This stacking interaction stabilises the dark CarH conformation that is conducive to the formation of head-to-tail dimeric interactions, which promotes tetramerisation[13]. This finding provides a structural rationale why *Tt*CarH is only capable of forming a stable tetramer when complexed with AdoCbl, whereas it remains a monomer when bound to OHCbl or methylcobalamin[3]. The cobalamin movement results in modest changes of conserved interactions at the monomer-monomer and dimer-dimer interface, and computation suggests this might weaken the tetrameric assembly. While crystals of the wild type quickly suffer from loss of diffraction upon prolonged light exposure, the *Tt*CBD-H132A variant does not lose lattice integrity upon aerobic in crystallo photoconversion, yielding a I$^{O2-H132A}$ structure containing a His177-Co(III)-H$_2$O species that resembles the wild type D$^{O2}$ structure with a water ligand replacing the Ado group. This further implies that the formation of the bis-His ligated cobalamin and particular His132 ligation is critical for tetramer disassembly/stabilisation of the monomer light-adapted state conformation, supported by solutions studies (Supplementary Figs. 1, 2a–h, 13a, b).

We thus propose that only the AdoCbl-bound dark state forms a tetramer (Fig. 9a, Supplementary Movie 1) by virtue of presence of the Ado ligand. Following photocleavage of the Co-C bond leading to Co(I) formation and exit of the adenosyl moiety, displacement of the corrin ring is accompanied by a ligand switch from lower axial His177 to upper axial His132, concomitant with the formation of the Co(II) species under anaerobic conditions. Following O$_2$-mediated oxidation of the Co(II)-His132 species, the formation of the Co(III) bis-His ligated corrin (via Co(III)-superoxo state and H$_2$O-Co(III)-H132 states) is achieved by relative domain reorientation concomitant with disintegration of the tetrameric assembly, a conformational change that is incompatible with crystal integrity. Under fully aerobic conditions, accumulation of a Co(II) species is not detected during oxidation of the transient Co(I) state (Fig. 3). Instead, an intermediate Co(III) species spectrally distinct from the final His177-Co(III)-H132 is observed to accumulate. Freeze-trapping following in crystallo photoconversion suggest this intermediate (I$^{O2}$) corresponds to an H$_2$O-Co(III)-H132 species, highly similar to the pentacoordinate Co(II)-His132 (I$^{anaer}$) readily observed under anaerobic conditions.

The results reported here provide unprecedented insights into the photo-induced structural changes of cobalamin in the B$_{12}$-dependent CarH family of photoreceptors. These results pave the way for future time-resolved serial crystallography experiments on these targets[30,31] at synchrotrons and XFELs to gain further structural insights into the photochemical mechanism of these unique photoreceptors.

## Methods

### Cloning, expression and purification

The genes for *Tt*CBD (residues 78–285, gene id WP_011174503.1) and its H132A and H177A variant were synthesised (GeneArt) and cloned in

pET21a (ampicillin resistant) with a C-terminal 6xHistag. A detailed protocol for expression and purification for full-length CarH[9], which has been published earlier, was adapted for the truncated *Tt*CBD and its variants. Briefly, genes were transformed in *E. coli* BL21(DE3) (New England Biolabs) cells and fresh transformants were used to grow cultures in sterile auto-induction media (Formedium) supplied with 50 μg/ml of ampicillin (Formedium). In a typical purification experiment cells from 2 l culture were resuspended in buffer containing 50 mM Tris-HCl pH 7.5, 100 mM NaCl and 10 mM imidazole. A two-step purification protocol, consisting of an affinity chromatography step followed by size-exclusion chromatography, similar to that of full-length *Tt*CarH was followed. *Tt*CBD also eluted as a tetramer from the Superdex Hiload 26/60 gel-filtration column (Cytiva) with >95%purity in elution buffer containing 50 mM Tris-HCl pH 7.5, 100 mM NaCl. Protein concentration was determined using published molar extinction coefficient of $8.0 \times 10^3 \, M^{-1} \, cm^{-1}$ for AdoCbl at 522 nm[32].

## SEC-MALS protocol

Size-exclusion chromatography coupled with multi-angle light scattering was carried out to measure the accurate molecular weight and oligomerisation state of the proteins in dark and light conditions. An Agilent G7110B HPLC pump, degasser and autoinjector (Agilent, Santa Clara, USA) was used to auto load the samples (50 μL each run, 1 mg/ml concentration). Superdex 200 10/300 GL column was used for chromatographic separations. MiniDAWN TREOS MALS detector, and Optilab rEX refractive index metre (Wyatt, Santa Barbara, USA) were used to collect light scattering signals. Flow rate was set at 1 mL min$^{-1}$ with mobile phase of pH 7.5 50 mM Tris-HCl, 100 mM NaCl buffer. All results were processed according to referenced protocol[33]. Peak alignment, band-broadening correction and normalisation procedures were performed by selecting the central 50% region of the peaks.

## Native mass spectrometry of truncated and full-length proteins

On the day of analysis, protein was desalted in 200 mM ammonium acetate (pH 7.0 for CBD domains or pH 8.0 for full-length [FL] proteins) using Micro Bio-Spin Chromatography columns (Bio-Rad, Micro Bio-Spin 6 Columns), following the manufacturer's instructions. Native MS data were acquired using the Thermo Scientific™ Q Exactive™ Hybrid Quadrupole-Orbitrap™ mass spectrometer (Thermo Fisher Scientific, Cramlington, UK). NanoESI capillaries were prepared in house from thin-walled borosilicate capillaries (inner diameter 0.9 mm, outer diameter 1.2 mm) (World Precision Instruments, Stevenage, UK) using a Flaming/Brown P-97 micropipette puller (Sutter Instrument Company, Novato, CA). Around 10 μL of 5 μM protein was used per capillary tube. A positive voltage was applied to the solution through a platinum wire (Goodfellow Cambridge Ltd., Huntington, UK) inserted into the capillary. For all spectra generated the spray current was kept between 0.2 and 0.3 μA with the spray voltage varying between 0.9–1.1 kV accordingly. The capillary temperature was kept constant at 350 °C and the S-lens RF level at 200. The resolution used was 25,000 for WT and 12,500 for H132A (due to its instability) with 10 microscans and an AGC target of 1e$^6$. The fore and ultra-high vacuums were kept constant at 1.58e mbar and 2.15e$^{-10}$ mbar respectively. No fragmentation was used and so HCD and NCE values were set to 0. WT protein was exposed to ambient light for 2 min at room temperature. H132A protein was exposed to ambient light and white LEDs (part of the Q Exactive™ system) for one hour at room temperature. Data was acquired for 5 min per condition. For analysis all 5 min were averaged and processed in Thermo Xcalibur (Thermo Fisher Scientific, Cramlington, UK).

## Size-exclusion chromatography of full-length proteins

A Superdex 200 10/300 GL column (Cytiva) was used to determine the oligomeric state of dark and light-adapted samples of full-length WT *Tt*CarH and its H132A variant. The experiment was performed under conditions identical to the one used in native mass spectrometry. All the runs were performed using an FPLC system at room temperature. A 200 μl aliquot of a 3 mg/ml (~100 μM) sample was injected into the column equilibrated in 200 mM ammonium acetate pH 8.0 buffer and eluted at a constant flow rate of 0.4 ml/min. For light-adapted samples an aliquot was exposed to light for 2 mins and injected on the column. Fractions comprising the tetrameric and monomeric fractions were pooled separately and a UV-Vis absorbance spectrum was recorded to confirm if the samples have converted to light state.

## Photoconversion followed using UV-Vis spectroscopy

A Cary 60 UV-Vis spectrophotometer (Agilent Technologies) was used to record spectra of dark- and light-adapted wild-type and variant *Tt*CBD samples. Around 25 μM dark-adapted samples (Tris-HCl buffer pH 7.5) in 1 cm pathlength quartz cuvettes were used to record absorbance spectra in the 300–800 nm region. To convert the sample to a light-adapted state, samples were exposed to a high-power 530 nm wavelength LED (M530L3, Thorlabs) for 15 secs.

For preparation of anaerobic samples, concentrated protein samples (250 μM) were moved to an anaerobic glovebox (Belle Technology) maintained in 100% $N_2$ environment at 21 °C and allowed to de-gas for 16 h. Buffers were de-gassed and purged using $N_2$ gas before being moved inside the glove box and left to further de-gas for 24 hrs. Dilution of concentrated samples (1:10) with purged and de-gassed buffer was done to achieve a 25 μM dark-adapted sample concentration. The diluted sample was transferred to a quartz cuvette and sealed using a rubber super seal. All the steps were performed inside the glove box under red light to avoid any unwanted photoconversion of the samples. Sealed cuvettes were carefully removed from the glove box and spectra was measured in the Cary 60 spectrophotometer. The anaerobic samples were illuminated using an identical setup as described above for aerobic samples. Spectra were recorded every minute to monitor the degradation of Co(I) cobalamin and the emergence of Co(II) cobalamin species. To convert the Co(II) cobalamin species to light-adapted bis-His ligated state, the super seals were removed and the samples were allowed to equilibrate with ambient $O_2$ and spectra were recorded every 5 min.

## Laser flash photolysis

Laser photoexcitation experiments were carried out at 25 °C using an Edinburgh Instruments LP980 transient absorption spectrometer. Samples contained *Tt*CBD in Tris-HCl buffer pH 7.5 and $O_2$ was removed by incubation in an anaerobic glovebox (Belle Technology) for 16 h. Photolysis was initiated by excitation at 530 nm (-15 mJ), using the optical parametric oscillator of a Q-switched Nd-YAG laser (NT342B, EKSPLA) in a cuvette of 1 cm pathlength. Kinetic transients at 388 nm were recorded with the detection system (comprising probe light, sample, monochromator, and photomultiplier tube detector) at right angles to the incident laser beam. Lifetimes were measured by fitting to a single exponential function using Origin Pro 9.1 software.

## Stopped-flow experiments

Stopped-flow kinetic measurements were carried out using an SX20 rapid mixing stopped-flow spectrophotometer (Applied Photophysics Ltd, Leatherhead, UK) placed inside a Belle Technology anaerobic chamber (oxygen levels <2 ppm). Data were collected at 25 ˚C using a photodiode array (PDA) detector. Anaerobically photolysed samples (CoI) of -50 μM WT *Tt*CBD (final concentration in the measurement cell) in Tris-HCl buffer was mixed against oxygenated buffer. Experiments were also repeated with anaerobically photolysed samples that had been allowed to age for ~2 h prior to mixing. Lifetimes were measured by fitting kinetic transients at the stated wavelength to a single or double exponential function using Origin Pro 9.1 software.

## Sample preparation for EPR spectroscopy

All EPR measurements were performed on samples of 300 μM concentration. The protein samples were allowed to de-gas for 16 h in the glove box and carefully transferred to EPR tubes under red light. Each tube holding ~200 μl of sample was sealed using rubber subaseal and parafilm to maintain them in $O_2$ free environment. The tubes were removed from the glove box and one dark sample was immediately flash-frozen in liquid $N_2$. All the other tubes were illuminated using the high-power 530 nm LED for 15 secs. To monitor the formation of paramagnetic Co(II) species in time an EPR tube was flash-frozen in liquid $N_2$ after 0, 2, 5, 10, 30 and 60 min of light illumination. The samples were stored at 77 K until further measurements were performed. To detect the conversion of Co(II) to Co(III)-superoxo species, the light-exposed 60 min anaerobic sample was annealed at room temperature for 2, 5 and 15 min to allow ambient $O_2$ to exchange with the sample in the tube, and EPR spectra was recorded at 20 K.

## EPR spectroscopy data collection and analysis software

All EPR samples were prepared in Tris-HCl buffer pH 7.5. Samples containing ~300 μM of wild-type *Tt*CBD and *Tt*CBD-H132A were transferred into 4 mm outer diameter/3 mm inner diameter Suprasil quartz EPR tubes (Wilmad LabGlass) and frozen in liquid $N_2$. The photoactivation of the *Tt*CBD and TtCBD-H132A was carried out at room temperature under anaerobic conditions. Optical irradiation at 530 nm (500 mW) was accomplished (for the specified duration described in the text/legend) using a Thorlabs Mounted High Power LED (M530L3) with the output beam collimated using a Thorlabs collimation adaptor (SM2F32-A). All EPR samples were measured on a Bruker EMXplus EPR spectrometer equipped with a Bruker ER 4112SHQ X-band resonator. Sample cooling was achieved using a Bruker Stinger[34] cryogen-free system mated to an Oxford Instruments ESR900 cryostat, and temperature was controlled using an Oxford Instruments MercuryITC. The optimum conditions used for recording the spectra are given below; microwave power 30 dB (0.22 mW), modulation amplitude 5 G, time constant 82 ms, conversion time 25 ms, sweep time 90 s, receiver gain 30 dB and an average microwave frequency of 9.383 GHz. All EPR spectra were measured as a frozen solution at 20 K, respectively. The analysis of the continuous wave EPR spectra were performed using EasySpin toolbox (5.2.28) for the Matlab program package[35].

## Crystallisation screening

**Form 1.** Only fractions comprising the top half of the elution peak from the gel-filtration column were used for screening of new crystallisation conditions. Pooled fractions for both purified wild-type *Tt*CBD and H132A samples were concentrated at 20 mg/ml. Using a nanolitre dispensing robot (Mosquito, SPT Labtech, UK), screening for new conditions was carried out with various crystallisation screens (Molecular Dimensions). Drops of 400 nl (200 nl protein + 200 nl reservoir) size were setup in 3 lens crystallisation plates (SWISSCI) under red light. The plates were left to incubate at 21 °C in opaque boxes. Crystals appeared in several conditions after overnight growth. Condition B3 from Morpheus screen (0.09 M halogens, 0.1 M buffer system 1 pH 6.5, 30% v/v precipitant mix 3) and condition F2 from LMB crystallisation screen (0.1 M ammonium sulfate, 0.1 M sodium citrate pH 5.8, 16% w/v PEG 4000, 20% v/v glycerol) gave best diffracting crystals for wild type *Tt*CBD and H132A variant respectively. Prior to flash freezing, crystals were transferred to a drop containing mother liquor supplemented with 20%w/v PEG 200 for cryo protection. All the crystal handling was done under red light to ensure crystals do not undergo photolysis.

**Form 2.** Crystallisation of WT *Tt*CBD under aerobic conditions was carried out by the batch method, at room temperature (20 °C) and under red light conditions. 500 μl of freshly produced protein at 8 mg/ml was mixed with 500 μl of a solution containing 20% PEG 10000, HEPES 0.1 M pH 7.5. After two days of incubation, the formed crystals reached their final size of 10–20 × 300 × 100 μm³.

## In crystallo photoconversion in aerobic and anaerobic conditions

**Form 1.** The dark-adapted crystals were transferred to a drop containing cryo-protectant and the red-light filter on the microscope light source was removed. Wild-type *Tt*CBD crystals showed cracks on the surface when exposed to microscope light under aerobic conditions and those crystals did not diffract well. The dark crystals of wild-type *Tt*CBD when harvested from the cryoprotected solution in a nylon loop and exposed to a 530 nm wavelength LED (ThorLabs) light for a period of 1–5 s still diffracted and allowed the elucidation of the 1.8 Å model of the intermediate tetramer state which showed the His-ligation switch under aerobic conditions. The H132A mutant crystals were stable and withstood light exposure up to minutes, which allowed successful data collection and obtaining a model for light-adapted H132A mutant.

To obtain anaerobic structures for both wild type and H132A mutant, crystallisation plates containing fresh crystals were moved inside the glove box and allowed to de-gas for 4 days. The plates were stored in opaque boxes to prevent photoactivation of crystals. All crystal handling and flash-cooling was performed inside the glove box under red light. Drops containing the crystals were exposed to white light from the microscope light source for 30–40 secs and the colour change from pink (dark) to orange (characteristic of Co(II) species) in the crystals was monitored. In the case of the H132A mutant crystals, the pink colour of the dark-adapted state turned pale initially which gradually turned orange on exposure to light for longer periods. The crystals turning pale point to the Co(I) state and then turning orange indicate the presence of Co(II) state that were trapped in crystallo. The moment these colour changes were visible the crystals were fished out and flash-cooled by plunging directly in liquid $N_2$. Crystals flash-cooled anaerobically were not cryoprotected.

**Form 2.** To generate a light-illuminated state, a WT *Tt*CBD crystal was cryoprotected in the crystallisation solution that additionally contained 15% PEG 200, loop-mounted under red light conditions at RT and illuminated for 5 s using a 530-nm LED source coupled to an optical fibre (diameter of 200 μm; Thorlabs). Crystals were 1.6 mm away from the fibre end, at a position where the output power was measured to be around 5 mW. When the illumination was carried out under white light conditions, a change in crystal colour from pink to orange was observed. Once the illumination was terminated, the crystal was flash-cooled in LN₂. As a dark-state control, a non-illuminated WT *Tt*CBD crystal was cryoprotected, loop-mounted and flash-cooled in LN₂, all under red-light conditions.

## X-ray data collection, structure refinement and analysis

**Form 1.** Unattended data was collected on beamlines i03 and i04 at Diamond Light Source using standard data collection parameters (100 K, 360° sweep, 0.1° oscillation). An average diffraction-weighted dose of 8.6 MGy was computed for individual datasets of form 1 crystals. Datasets were integrated, scaled and merged using the automated xia2dials[36] pipeline available on the ISPyB web user interface. Both wild type and H132A mutant of *Tt*CBD crystallised in the space group $P2_12_12_1$ with a complete tetramer in the asymmetric unit. Monomeric truncated model of *Tt*CarH (PBD 5C8A)[12] was used as the search model to find a complete tetramer in the 1.8 Å dark-adapted *Tt*CBD model ($a = 64.1$, $b = 69.7$, $c = 204.5$, 90.0°, 90.0°, 90.0°). Subsequently, this high-resolution model (PDB 8C31) was used as a search model for the H132A mutant and for the light-exposed models. Structures were refined using refmac5[37] from the ccp4i2 software suite[38]. Refinement with TLS gave better results so automatically

defined TLS parameters were used during refinement cycles. Data collection and refinement statistics are reported in Supplementary Table 1. PDBredo was also used early during refinement to improve the quality of the models[39]. Manual model building was carried out in COOT[40] and Pymol 2.5 (Schrodinger) was used for analysis of refined models and for preparing figures.

**Form 2.** X-ray crystallographic data were collected at 100 K on dark-state and illuminated WT *Tt*CBD crystals at the ID23-1 beamline[41] of the ESRF, using a photon energy of 14.0 keV. 180° sweep, 0.1° oscillation. On each crystal, 1800 images were collected with a 0.1° oscillation range and an exposure time of 30 ms each, using an Eiger2 16 M detector. The X-ray beam was 30 μm (h) × 30 μm (v) FWHM in size and carried a photon flux of $5.34 \times 10^{11}$ ph/sec. An average diffraction-weighted dose of 9.3 MGy per data set was calculated using Raddose-3D[42]. Diffraction intensities were integrated, scaled and merged using the automated data reduction pipeline[43] used at the ESRF and the MTZ files downloaded from the ISPyB database. Upon light-illumination, the *a* axis shortened by 2.7 Å and the *b* axis lengthened by 6.6 Å (see Supplementary Table 2). A monomeric truncated model of *Tt*CarH (PDB 5C8A)[12] was used as a search model. The space group was $P2_12_12_1$, with one tetramer per asymmetric unit. Structures were refined in reciprocal space using *PHENIX*[44] and real space refinement was carried out using *COOT*[40].

### RMSD calculations and superpositions
Calculation of Cα-backbone rmsd values was performed using the PDBefold (European Bioinformatics Institute) server[45]. For superpositions LSQKAB[45,46] from ccp4 software suite was used. The wild-type dark structure was used as a fixed reference structure. Residues 83 to 265 from each chain in illuminated structures were superimposed on corresponding residues in the dark structure and displacement for all Cα atoms was determined. The original b-factor column in the illuminated structures was replaced with the displacement values obtained from lsqkab and the script loadbfacts.py (Gatti-Lafranconi, P. Pymol script: loadBfacts.py.https://doi.org/10.6084/m9.figshare.1176991.v1(2014)) was used to generate a figure in Pymol.

### Electrostatic potential calculations
For the calculation of electrostatic potentials, PDB files were first prepared so that the dark and illuminated structures contained the same atoms, to ensure that calculated differences are due entirely to structural changes; to this end, every atom not present in both structures was deleted. Electrostatic forces and binding energies were then calculated using DelphiForce[29] using PQR files created from crystal structure PDB files using PDB2PQR version 2.1.0[47] at pH 7.0. For the electrostatic potential maps, potentials were calculated using the APBS (Adaptive Poisson-Boltzmann Solver) server[48] and the maps were generated using USCF ChimeraX version 1.3[49].

### Reporting summary
Further information on research design is available in the Nature Portfolio Reporting Summary linked to this article.

## Data availability
PDB coordinates for the structures reported in this study have been deposited in the Protein Data Bank with the accession codes 5C8A, 5C8D, 5C8E, 5C8F, 8C31, 8C32, 8C33, 8C34, 8C35, 8C36, 8C37, 8C73 and 8C76. All other source data are provided as supplementary data files. Source data are provided in this paper.

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

## Acknowledgements

The authors would like to thank the Diamond Light Source (proposal mx24447) and the ESRF (proposal MX-2398) for beamtime, and the staff of beamlines I03, I04 and ID23-1 for assistance with data collection. We thank the EPSRC for access to the National Research Facility for Electron Paramagnetic Resonance Spectroscopy (NS/A000055/1, EP/W014521/1). Funding from an EPSRC International Centre-to-Centre grant no EP/S030336/1 awarded to D.J.H., D.L. and N.S.S. and from an ANR grant (PhotoGene, ANR-21-CE11-0036-01) awarded to G.S. supported the research reported in this paper. R.R.S. thanks the CEA for a CFR Ph.D. fellowship. This work was partially carried out at the platforms of the Grenoble Instruct-ERIC center (IBS and ISBG; UMS 3518 CNRS-CEA-UGA-EMBL) within the Grenoble Partnership for Structural Biology (PSB). Platform access was supported by FRISBI (ANR-10-INBS-05-02) and GRAL, a project of the University Grenoble Alpes graduate school (Ecoles Universitaires de Recherche) CBH-EUR-GS (ANR-17-EURE-0003). The IBS acknowledges integration into the Interdisciplinary Research Institute of Grenoble (IRIG, CEA).

## Author contributions

H.P. carried out the expression and purification of all proteins with assistance from M. Sakuma. H.P. and D.J.H. designed the spectroscopy experiments, collected data and performed analysis. H.P. prepared samples for EPR spectroscopy and M. Shanmugam and A.B. collected and analysed the data. H.P. performed crystallisation, crystal freezing and structure determination for form 1 models reported in the paper. C.L. helped with X-ray data collection and producing the supplementary movie. L.N.J. performed native mass spectrometry on different samples. S.Z. ran the SEC-MALS experiments and analysed the data. L.J. and S.H. carried out the computational calculations. H.P. and D.L. devised the crystallisation and illumination strategies, and refined, analysed and interpreted the form 1 crystal structures. Crystallogenesis and X-ray data collection of form 2 crystals were carried out by R.R.S. The illumination protocol to generate the light-adapted state has been established and applied to form two crystals by G.S. and R.R.S. Structure refinement of form 2 crystals has been carried out by R.R.S. and J.P.C. H.P., D.J.H., N.S.S. and D.L. planned the work together with M.W. and wrote the manuscript with inputs from all the authors.

## Competing interests

The authors declare no competing interests.

 
