## [Peer Review File · Nature Communications]

REVIEWER COMMENTS

Reviewer #1 (Remarks to the Author):

The paper identifies for the first time intermediate states indicating photo-induced structural changes of cobalamin in the B12-dependent CarH family of photoreceptors. The work is highly significant in the field and provides unique insights important for a number of applications. The work provides convincing structural and spectroscopic data that fully supports the conclusions made with analysis of the WT and H132A to confirm the role of His132 ligation.

The methodology used is well established, in particular the use of cryo-trapping crystallography is well suited to the time scale of processes under investigation. I do suggest that a table 1 is included in the main paper to allow quick comparison of structure quality and some indication of the diffraction weighted dose should be included. I look forwards to seeing results of future time-resolved serial crystallographic studies.

The methods are complete and allow reproduction of the work.

Reviewer #2 (Remarks to the Author):

This submission reports a combined spectroscopic / crystallographic study of the B12-dependent photoreceptor, CarH. This is an interesting system, being the first studied example of a light-activated protein that exploits B12 photochemistry. As such there are a number of outstanding questions regarding the photoactivation mechanism. The authors rightly point out that not all of the photochemical intermediates are well understood, nor is the role of oxygen, and the full extent of the structural dynamics that follow photoactivation are still to be elucidated.

The data presented provide some useful insights: the likely Co(I) intermediate, the dissociation of the lower axial H177, and the change in interactions that stabilise the tetramer. The most compelling data, however, are those acquired under anaerobic (i.e., non-native) conditions. Here, they have direct evidence of certain intermediates (e.g., Co(II), using EPR) and it is interesting that subsequent exposure to O₂ leads the system to the final photoproduct. The data are more limited for the aerobic case, however, and I fear the authors have made some conclusions about the native mechanism that their data do not support (detailed below).

Consequently, although promising, this feels like a work in progress. I don't think the new insights listed above are sufficient for publication in Nature Communications, and more data are required to support some of their other conclusions. For this reason, alongside some other experimental concerns I outline below, I cannot support publication at this stage.

Major points

1. The data in Figure S1 are cited in the main manuscript as evidence of WT and H132A TtCBD forming stable tetramers in the dark. The estimated molecular weights of each varies by almost 20 kDa (75.6 and 93.4 kDa, respectively), however, which doesn't appear to be mentioned. The H132A data appear to make sense, but why is the WT domain eluting so small? From the estimated MW of the WT light monomer state, 23.6 kDa, are the WT tetramer data not more consistent with a trimer, or is this a typographical error?

2. I'm having trouble following how the data are presented in Figures 2b and S2a. I agree that the red spectrum in each likely corresponds with a Co(I) species, but why would this be formed following 0 mins illumination as indicated in each figure? Surely photolytic cleavage of the Ado is required to form this state, or have I missed something?

3. Although they mention that it does, they offer no explanation for why they think the buffer makes a difference to the Co(I) lifetime?

4a. On page 13 (lines 266-268), the authors state "The increase in His132-Co bond length likely correlates with a change from pentacoordinate Co(II) to the hexacoordinate Co(III) state upon oxidation." As written, this could be read to imply that Co(II) is a productive intermediate in the aerobic mechanism, which gets oxidised to a hexacoordinate Co(III) state in the presence of oxygen. If that is the implication, I don't think they have any direct evidence for this. I don't necessarily doubt the H132-Co bond length could be indicative of penta- vs hexacoordinate, but the comparison is between two different sets of conditions (anaerobic vs aerobic) and not a change from Co(II) to Co(III) within a given set of conditions. If this isn't what is being implied, then this statement should be reworded for greater clarity.

4b. This gets to my main concern about how they have interpreted their data. They only have direct evidence, by EPR, of the Co(II) intermediate for data acquired under anaerobic conditions. Although it is interesting that this Co(II) state generated in the absence of O₂ can then convert to the bis-His adduct light state following the subsequent introduction of O₂, this is not the same as saying that Co(II) is an intermediate in the fully aerobic mechanism, as appears to be the suggestion in Figure 8. As things stand they do not have the evidence for this. The data presented do not have sufficient time resolution to

have captured all of the intermediates involving oxygen. Are they set up to do time-resolved EPR measurements at ambient temperature as they would be the ideal measurement to capture the putative Co(II)?

5. On Page 14, the authors acknowledge that the full-length H132A variant studied in ref 12 undergoes light-triggered tetramer disassembly, in contrast to their current observations that the TtCBD-H132A mostly does not. The following statement in ref 12: “Notably, both WT and H132A CarH undergo light-dependent tetramer disassembly, indicating that bis-His ligation is not required...” was based on both spectroscopic and SEC data, confirming 4mer to 1mer transition. This directly contradicts a central conclusion of this manuscript; i.e., that conversion to Co(III) in the absence of H132 is not sufficient to lead to tetramer dissociation. Does this not undermine the relevance of the TtCBD data to what happens in the full-length protein? To investigate this, the authors need to compare, more fully, the behaviour of the TtCBD-H132A with the full-length H132A in their own hands, preferably by UV-vis and SEC as was carried out for the full-length version in ref 12.

6. A general point is that where the figure panels are referred to (or not) is a bit all over the place and makes following the paper challenging. Sometimes it appears to be wrong, in other places observations are referred to, but with no indication of the Figure the reader should be looking at. I point to some examples below, but the entire manuscript requires editing with this fact in mind.

More minor points

Page 1, lines 31-33: I’m unsure why the authors specify “thermostable bacteria”. Although much of the work on isolated proteins has been conducted on CarH from *Thermus thermophilus*, this class of protein is not exclusive to thermophilic bacteria. CarH was first discovered in the gram-negative bacterium *Myxococcus xanthus*, homologous sequences have been found in the genomes of several other strains, and CarH-like proteins are thought to be widespread in non-photosynthetic bacteria. If the authors mean something else by the word thermostable, could they please elaborate; otherwise, I suggest they edit this accordingly.

Page 3, lines 70-71. The authors point out that important insights have been provided by transient absorption spectroscopy. For the more general reader, perhaps they can briefly summarise in the Introduction what these insights are.

Page 4, Lines 86-87. The statement “...the formation of Co(III) does not result in tetramer dissociation in the H132A variant.” is vague and its significance isn’t very clear to general readers without prior knowledge of the system.

Page 9, line 181. "As a consequence, the pentacoordinate Co atom is ligated by His132, which has moved inwards to form the new His132-Co bond (Figure S7a)." Presumably the authors mean to refer to Fig 3c not Fig S7a?

Page 12, lines 229-230. "...and reported to be important for dimer-dimer contact...". If reported elsewhere, surely the paper(s) should be cited?

Page 14, lines 298-300. "...with complete conversion to Co(II) species observed in ~120 min under anaerobic conditions at room temperature (Figure S10c)." As far as I can tell, whereas the data acquired in Tris in Figure 6a suggest complete conversion to Co(II) in 120 mins, the data in Figure S10c indicate the conversion is significantly slower in phosphate buffer? If so, this clause needs to be edited to reflect this.

Page 30, lines 695-696: The title of ref 22 is incomplete.

Figure S3. It should be made clear in the Figure legend that these data were collected under anaerobic conditions

Reviewer #3 (Remarks to the Author):

I just comment on the cw-EPR characterization of the Co(II) and the Co(III) (superoxo and light-adapted) states. The EPR spectra and their assignment seems reasonable. In the methods section analysis of the spectra by EasySpin is described, but neither in the manuscript nor in the SI any quantitative values (g-tensors, hf couplings) nor simulations of the spectra are shown. These values should be given and compared to published values for these species.

RESPONSE TO REVIEWER COMMENTS

Reviewer 1:

The paper identifies for the first time intermediate states indicating photo-induced structural changes of cobalamin in the B12-dependent CarH family of photoreceptors. The work is highly significant in the field and provides unique insights important for a number of applications. The work provides convincing structural and spectroscopic data that fully supports the conclusions made with analysis of the WT and H132A to confirm the role of His132 ligation.

The methodology used is well established, in particular the use of cryo-trapping crystallography is well suited to the time scale of processes under investigation. I do suggest that a table 1 is included in the main paper to allow quick comparison of structure quality and some indication of the diffraction weighted dose should be included. I look forwards to seeing results of future time-resolved serial crystallographic studies.

The methods are complete and allow reproduction of the work.

We would like to thank the referee for their positive comments on our manuscript. As suggested, we have moved the data collection and refinement statistics Tables S3 and S4 from the SI to the main text, now labelled Table 1 and Table 2. We have also added a line in the methods section indicating the average diffraction weighted dose for the datasets.

Reviewer 2:

This submission reports a combined spectroscopic / crystallographic study of the B12-dependent photoreceptor, CarH. This is an interesting system, being the first studied example of a light-activated protein that exploits B12 photochemistry. As such there are a number of outstanding questions regarding the photoactivation mechanism. The authors rightly point out that not all of the photochemical intermediates are well understood, nor is the role of oxygen, and the full extent of the structural dynamics that follow photoactivation are still to be elucidated.

The data presented provide some useful insights: the likely Co(I) intermediate, the dissociation of the lower axial H177, and the change in interactions that stabilise the tetramer. The most compelling data, however, are those acquired under anaerobic (i.e., non-native) conditions. Here, they have direct evidence of certain intermediates (e.g., Co(II), using EPR) and it is interesting that subsequent exposure to O₂ leads the system to the final photoproduct. The data are more limited for the aerobic case, however, and I fear the authors have made some conclusions about the native mechanism that their data do not support (detailed below).

Consequently, although promising, this feels like a work in progress. I don't think the new insights listed above are sufficient for publication in Nature Communications, and more data are required to support some of their other conclusions. For this reason, alongside some other experimental concerns I outline below, I cannot support publication at this stage.

We thank this reviewer for their positive comments about the importance and relevance of our work, and also the meticulous critique of our studies. Below, we outline the responses to the points raised:

Major points

1. *The data in Figure S1 are cited in the main manuscript as evidence of WT and H132A TtCBD forming stable tetramers in the dark. The estimated molecular weights of each varies by almost 20 kDa (75.6 and 93.4 kDa, respectively), however, which doesn't appear to be mentioned. The H132A data appear to make sense, but why is the WT domain eluting so small? From the estimated MW of the WT light monomer state, 23.6 kDa, are the WT tetramer data not more consistent with a trimer, or is this a typographical error?*

We understand the concern raised by the reviewer and we found this puzzling too. Given the retention times for both dark WT and H132A variant of TtCBD are almost identical we assumed the discrepancy in mass was due to the light scattering equipment. However, to further probe the oligomeric state of dark WT TtCBD we performed native mass spectrometry on dark and light adapted samples of WT TtCBD and the H132A variant. The results of these experiments clearly point to the existence of an oligomeric dark state with mass consistent with the tetrameric weight of the truncated proteins. Light adapted sample of WT TtCBD showed monomeric species whereas the H132A variant showed a large population of light adapted tetramer with minor dimeric and a higher oligomeric state which roughly corresponds to the mass of 2 tetramers. These data have been added in the supplementary information as Figure S2 just after the SEC-MALS data.

2. *I'm having trouble following how the data are presented in Figures 2b and S2a. I agree that the red spectrum in each likely corresponds with a Co(I) species, but why would this be formed following 0 mins illumination as indicated in each figure? Surely photolytic cleavage of the Ado is required to form this state, or have I missed something?*

We apologise for the confusion regarding this figure and the reviewer is correct to point out that the Co(I) species only forms following photolytic cleavage. The times indicated in the figure refer to the length of time the sample has been incubated anaerobically in the dark following illumination for 15 s. Hence, the '0 min' sample corresponds to sample that has been illuminated for 15 s and the spectra recorded straight away. We clarified this more clearly in the revised version, by including "*The times indicated in the key refer to the length of time samples were incubated in the dark following illumination for 15s.*"

3. *Although they mention that it does, they offer no explanation for why they think the buffer makes a difference to the Co(I) lifetime?*

Although we are not fully sure why the Co(I) lifetime is dependent on the nature of the buffer system, it is clear that the Co(I) state is highly reactive and likely to react with certain buffer components. We have added the following sentence to explain the change in lifetime in phosphate buffer: '*The lifetime of the Co(I) cobalamin species is sensitive to the buffer system used and increases in phosphate buffer, likely due to more inert nature of phosphate buffer system vis-a-vis Tris buffer.*'

4a. *On page 13 (lines 266-268), the authors state "The increase in His132-Co bond length likely correlates with a change from pentacoordinate Co(II) to the hexacoordinate Co(III) state upon oxidation." As written, this could be read to imply that Co(II) is a productive intermediate*

in the aerobic mechanism, which gets oxidised to a hexacoordinate Co(III) state in the presence of oxygen. If that is the implication, I don't think they have any direct evidence for this. I don't necessarily doubt the H132-Co bond length could be indicative of penta- vs hexacoordinate, but the comparison is between two different sets of conditions (anaerobic vs aerobic) and not a change from Co(II) to Co(III) within a given set of conditions. If this isn't what is being implied, then this statement should be reworded for greater clarity.

We agree with the referee that the structural changes do not provide any direct evidence that that Co(II) is a productive intermediate in the aerobic mechanism. Consequently, we have removed this statement in revision, and have changed this section accordingly in light of the additional experiments described below. We have modified Fig. 8 in a way that Co(II) does not appear as a productive intermediate under aerobic conditions. We thank the referee for the insight he/she provided that allowed clarifying the mechanism as summarized in Fig. 8.

4b. This gets to my main concern about how they have interpreted their data. They only have direct evidence, by EPR, of the Co(II) intermediate for data acquired under anaerobic conditions. Although it is interesting that this Co(II) state generated in the absence of O₂ can then convert to the bis-His adduct light state following the subsequent introduction of O₂, this is not the same as saying that Co(II) is an intermediate in the fully aerobic mechanism, as appears to be the suggestion in Figure 8. As things stand they do not have the evidence for this. The data presented do not have sufficient time resolution to have captured all of the intermediates involving oxygen. Are they set up to do time-resolved EPR measurements at ambient temperature as they would be the ideal measurement to capture the putative Co(II)?

We thank the reviewer for this insightful comment and have investigated this further during the revision process. Although the Co(II) state is observed under anaerobic conditions, our data clearly show that this is preceded by the formation of a Co(I) intermediate, which only slowly decays to the Co(II) state over the minutes-hours timescale. In the original manuscript we only showed that O₂ was able to react with this Co(II) state to form the bis-His light state, but as the reviewer correctly points out, in fully aerobic conditions it is likely that the O₂ would actually react with the Co(I) state. However, in previous time-resolved spectroscopy experiments by Kutta et al (ref 21) this Co(I) species was not observed.

To rationalise these findings, we have now used a combination of laser flash photolysis and stopped-flow spectroscopy measurements to measure the rates of formation / decay of the Co(I) intermediate (see Figure 3). Under anaerobic conditions, our new laser flash photolysis data show that the Co(I) intermediate forms with a lifetime of ~10 ms. Interestingly, if we then rapidly mix the anaerobically photolyzed protein (i.e. Co(I) state) with O₂ then this same intermediate decays with a lifetime of ~1 ms. Therefore, as the reaction of the Co(I) intermediate with O₂ is much faster than its rate of formation this explains why the Co(I) state is not observed under fully aerobic conditions in Kutta et al. We have also repeated this same stopped flow experiment with anaerobically photolyzed sample that has been allowed to age to the Co(II) for ~1 hour. In this case, the reaction with O₂ to form the light state is much slower (lifetime of >1 s). Hence, the referee is correct that the Co(II) state is unlikely to be an intermediate in the fully aerobic pathway. We have now clarified this in the revised manuscript and have changed the proposed mechanistic pathways shown in Figure 8 to include both the aerobic and anaerobic pathways. We point out that under physiological conditions both pathways could be operational where *in vivo* O₂ levels are limited.

5. On Page 14, the authors acknowledge that the full-length H132A variant studied in ref 12

undergoes light-triggered tetramer disassembly, in contrast to their current observations that the TtCBD-H132A mostly does not. The following statement in ref 12: “Notably, both WT and H132A CarH undergo light-dependent tetramer disassembly, indicating that bis-His ligation is not required...” was based on both spectroscopic and SEC data, confirming 4mer to 1mer transition. This directly contradicts a central conclusion of this manuscript; i.e., that conversion to Co(III) in the absence of H132 is not sufficient to lead to tetramer dissociation. Does this not undermine the relevance of the TtCBD data to what happens in the full-length protein? To investigate this, the authors need to compare, more fully, the behaviour of the TtCBD-H132A with the full-length H132A in their own hands, preferably by UV-vis and SEC as was carried out for the full-length version in ref 12.

As suggested, we have performed size exclusion chromatography (Superdex200 10/300 GL) on both WT and H132A variant of full length TtCarH. Our results with the full-length protein also show a trend identical to the truncated proteins. Full-length H132A variant predominantly remains a light adapted tetramer after light activation (see Figure S13). The fractions comprising the tetramer or monomer peaks were pooled and UV-Vis spectra were recorded to confirm the states of these samples. We further subjected full-length proteins to native mass spectrometry to further confirm the oligomeric integrity of full-length proteins (see Figure S2).

6. A general point is that where the figure panels are referred to (or not) is a bit all over the place and makes following the paper challenging. Sometimes it appears to be wrong, in other places observations are referred to, but with no indication of the Figure the reader should be looking at. I point to some examples below, but the entire manuscript requires editing with this fact in mind.

We apologise for any confusion with our referencing of figures in the original version of the manuscript. We clarified this in the revision.

More minor points

Page 1, lines 31-33: I'm unsure why the authors specify “thermostable bacteria”. Although much of the work on isolated proteins has been conducted on CarH from Thermus thermophilus, this class of protein is not exclusive to thermophilic bacteria. CarH was first discovered in the gram-negative bacterium Myxococcus xanthus, homologous sequences have been found in the genomes of several other strains, and CarH-like proteins are thought to be widespread in non-photosynthetic bacteria. If the authors mean something else by the word thermostable, could they please elaborate; otherwise, I suggest they edit this accordingly.

We have removed the term thermostable.

Page 3, lines 70-71. The authors point out that important insights have been provided by transient absorption spectroscopy. For the more general reader, perhaps they can briefly summarise in the Introduction what these insights are.

We have added the following sentences in the introduction section: ‘Kutta *et al* suggested that the Co-C bond scission is likely to proceed via a heterolytic cleavage route, leading to further distinct spectral intermediates on the μ s-ms timescales and tetramer dissociation over several seconds²¹. However, Miller and co-workers have proposed the formation of a long-lived triplet state upon excitation, which ultimately reacts to form the stable 4', 5' anhydroadenosine product and final light state of CarH²².’

Page 4, Lines 86-87. The statement "...the formation of Co(III) does not result in tetramer dissociation in the H132A variant." is vague and its significance isn't very clear to general readers without prior knowledge of the system.

We have clarified this in the revised version and it now reads as follows: 'While Co oxidation ultimately drives monomer formation and bis-His ligation in the WT in solution, tetramer dissociation is also dependent on the presence of the upper His ligand and does not occur in a H132A variant.'

Page 9, line 181. "As a consequence, the pentacoordinate Co atom is ligated by His132, which has moved inwards to form the new His132-Co bond (Figure S7a)." Presumably the authors mean to refer to Fig 3c not Fig S7a?

We apologise for this oversight and have corrected in the revised version.

Page 12, lines 229-230. "...and reported to be important for dimer-dimer contact...". If reported elsewhere, surely the paper(s) should be cited?

We have cited reference [12] in relation to this statement.

Page 14, lines 298-300. "...with complete conversion to Co(II) species observed in ~120 min under anaerobic conditions at room temperature (Figure S10c)." As far as I can tell, whereas the data acquired in Tris in Figure 6a suggest complete conversion to Co(II) in 120 mins, the data in Figure S10c indicate the conversion is significantly slower in phosphate buffer? If so, this clause needs to be edited to reflect this.

We have added this line in the associated section: "The rate of decay of Co(I) cobalamin species is significantly slower in phosphate buffer when compared to Tris buffer conditions (Figure S14c)."

Page 30, lines 695-696: The title of ref 22 is incomplete.

This has been corrected accordingly.

Miller, N. A. et al. The Photoactive Excited State of the B₁₂-based photoreceptor CarH. *J Phys Chem B* 124, 10732-701 10738, doi:10.1021/acs.jpcc.0c09428 (2020).

Figure S3. It should be made clear in the Figure legend that these data were collected under anaerobic conditions

This has now been added accordingly.

Reviewer 3:

I just comment on the cw-EPR characterizat on of the Co(II) and the Co(III) (superoxo and light-adapted) states. The EPR spectra and their assignment seems reasonable. In the methods section analysis of the spectra by EasySpin is described, but neither in the manuscript nor in

the SI any quantitative values (g-tensors, hf couplings) nor simulations of the spectra are shown. This values should be given and compared to published values for these species.

The authors thank this reviewer for their comments on the EPR results and as desired we have included the simulations and quantitative values for g-tensors and hyperfine couplings in the supplementary information of the manuscript. The new figure and associated legend in numbered as Fig S5.

Figure S5: Experimental cw-EPR spectra (black traces) of the Co(II) (bottom trace) and Co(III)-super-oxo (top trace) species of *Tt*CBD enzyme measured at 20 K as a frozen solution. The simulations (red traces) are overlaid for a comparison. The spin-Hamiltonian parameters used to model the EPR spectra are given below; Co(II)-species – $\mathbf{g} = [2.003 \ 2.216 \ 2.259]$, $\mathbf{A}^{(59\text{Co})} = [300 \ 20 \ 15]$ MHz, $\mathbf{A}^{(14\text{N})} = [52 \ 36 \ 52]$ MHz, line widths = [0.37 0.2] mT and HStrain= [20 110 100] MHz; Co(III)-super-oxo species – $\mathbf{g} = [1.998 \ 2.002 \ 2.075]$, $\mathbf{A}^{(59\text{Co})} = [17 \ 35 \ 42]$ MHz, line widths = [0.54 0.48] mT and HStrain= [0 0 15] MHz. The extracted \mathbf{g} - and hyperfine tensors agree with the reported values (references; *Coord. Chem. Rev.*, **1981**, 39, 295, *J. Am. Chem. Soc.*, **2012**, 134, 796, *J. Am. Chem. Soc.*, **2016**, 138, 14186, *J. Am. Chem. Soc.*, **2012**, 141, 10984, *Appl. Magn. Reson.*, **2001**, 20, 35).

REVIEWERS' COMMENTS

Reviewer #2 (Remarks to the Author):

The authors have addressed several of the points raised in my original review. In particular, the fact that they acknowledge that they don't yet have direct evidence of a Co(II) intermediate in the photoresponse of CarH under aerobic conditions, means some of their conclusions are now better supported by their data. Despite having claimed to have removed it in their rebuttal letter, however, the line "The increase in His132-Co bond length likely correlates with a change from pentacoordinate Co(II) to the hexacoordinate Co(III) state upon oxidation." remains in the manuscript in reference to the illuminate structure in the presence of oxygen (p18). I think it is important that this is removed or reworded, because it could give the false impression that evidence exists for a Co(II) intermediate in the aerobic mechanism

Similarly, in the Abstract: "Unexpectedly, in the absence of oxygen, Co-C bond cleavage is followed by reorientation of the corrin ring and a switch from a lower to upper histidine-Co ligation, corresponding to a pentacoordinate Co(II) state. Under aerobic conditions, rapid flash-cooling of crystals prior to deterioration upon illumination confirm a similar B12-ligand switch occurs." The way this passage is written, the reader could easily conclude that there is direct evidence in this manuscript for a Co(II) intermediate in the aerobic mechanism. To avoid this confusion, the authors should reword this.

Other issues remain, which are again related to conclusions drawn about the aerobic mechanism from data acquired under anaerobic, or partial anaerobic, conditions. From the new data, for example, it is interesting that the Co(I) intermediate reacts with O₂ with kinetics that are faster than those with which Co(I) is formed under anaerobic conditions. Although this provides a possible explanation for why Co(I) is not observed during previous transient absorption measurements of the native photorepsonse under aerobic conditions, this cannot be stated unequivocally. Much like for Co(II), they have no direct evidence of Co(I) in aerobic pathway – they only know that Co(I) is formed under anaerobic conditions and then reacts with O₂ upon subsequent introduction, and that the former happens with slower kinetics than the latter. These processes might be important in the native mechanism, but the data presented here aren't conclusive of this. Can the authors rule out that photoexcitation in presence of O₂ result in another mechanism that does not form a 5 coordinate Co(I)?

This potential conflation of mechanistic details under different conditions leaves a slightly confused picture, and my view remains that much of this is a work in progress. The ligation changes – dissociation of H177 and the importance of H132 – are the major findings of this paper that are supported directly and clearly by the data for the native mechanism. However interesting these may be, alone they probably don't advance the field sufficiently for publication in Nature Communications.

A few more minor points:

The native mass spec data in Fig S2 convincingly show that WT TtCBD forms a tetramer. I think it important, however, that the authors are more transparent in the main manuscript about the ambiguity in the SEC-MALS data in Fig S1. They should also reword the following sentence at the beginning of this section, which in light of the SEC-MALS data is misleading: “The AdoCbl-bound TtCBD showed identical light-activated spectroscopic behaviour to the full-length TtCarH protein.”

In contrast to ref 12, who apparently show using SEC and UV-visible absorption that under illumination the full length H132A 4mers convert mostly to 1mers, in Fig S13 the authors find only partial monomerization of this variant under illumination using similar measures (plus native mass spec in Fig S2). Could this partial photoconversion be owing to differences in illumination conditions (intensity, illumination time etc)? This should be discussed for the avoidance of doubt. Also, their evidence in Fig S14 that the light-adapted TtCBD-H132A has a spectrum more consistent with a hydroxylated Co(III) species than the bis-His adduct is convincing and an interesting finding. It's not clear to me from the data, however, if this species is formed exclusively in the truncated CBD protein or also in this variant for the full-length protein?

In response to my review, the authors added the following passage to the introduction: “Kutta et al suggested that the Co-C bond scission is likely to proceed via a heterolytic cleavage route, leading to further distinct spectral intermediates on the μs -ms timescales and tetramer dissociation over several seconds²¹. However, Miller and co-workers have proposed the formation of a long-lived triplet state upon excitation, which ultimately reacts to form the stable 4', 5' anhydroadenosine product and final light state of CarH²².” As written, this sounds like heterolytic bond breakage and a long-lived triplet state a mutually exclusive, which they aren't. They should make the case for why they think they are or edit this accordingly to remove this implication.

Reviewer 2:

The authors have addressed several of the points raised in my original review. In particular, the fact that they acknowledge that they don't yet have direct evidence of a Co(II) intermediate in the photoresponse of CarH under aerobic conditions, means some of their conclusions are now better supported by their data. Despite having claimed to have removed it in their rebuttal letter, however, the line "The increase in His132-Co bond length likely correlates with a change from pentacoordinate Co(II) to the hexacoordinate Co(III) state upon oxidation." remains in the manuscript in reference to the illuminate structure in the presence of oxygen (p18). I think it is important that this is removed or reworded, because it could give the false impression that evidence exists for a Co(II) intermediate in the aerobic mechanism

We apologize for this oversight and have now removed all mention of the Co redox state in this section. This no longer gives any false impression that evidence exists for a Co(II) intermediate in the aerobic mechanism. The new sentence reads "The increase in His132-Co bond length likely correlates with a change from pentacoordinate to the hexacoordinate state in the presence of oxygen."

Similarly, in the Abstract: "Unexpectedly, in the absence of oxygen, Co-C bond cleavage is followed by reorientation of the corrin ring and a switch from a lower to upper histidine-Co ligation, corresponding to a pentacoordinate Co(II) state. Under aerobic conditions, rapid flash-cooling of crystals prior to deterioration upon illumination confirm a similar B12-ligand switch occurs." The way this passage is written, the reader could easily conclude that there is direct evidence in this manuscript for a Co(II) intermediate in the aerobic mechanism. To avoid this confusion, the authors should reword this.

Again, we have removed any reference to the Co redox state in the abstract.

Other issues remain, which are again related to conclusions drawn about the aerobic mechanism from data acquired under anaerobic, or partial anaerobic, conditions. From the new data, for example, it is interesting that the Co(I) intermediate reacts with O₂ with kinetics that are faster than those with which Co(I) is formed under anaerobic conditions. Although this provides a possible explanation for why Co(I) is not observed during previous transient absorption measurements of the native photorepsonse under aerobic conditions, this cannot be stated unequivocally. Much like for Co(II), they have no direct evidence of Co(I) in aerobic pathway – they only know that Co(I) is formed under anaerobic conditions and then reacts with O₂ upon subsequent introduction, and that the former happens with slower kinetics than the latter. These processes might be important in the native mechanism, but the data presented here aren't conclusive of this. Can the authors rule out that photoexcitation in presence of O₂ result in another mechanism that does not form a 5 coordinate Co(I)?

We have added "However, it should be noted that alternative mechanisms where photoexcitation in the presence of O₂ does not result in the formation of a Co(I) state cannot be ruled out." to the end of **Role of Co(I) and Co(II) redox states upon aerobic and anaerobic photolysis results section.**

A few more minor points:

The native mass spec data in Fig S2 convincingly show that WT TtCBD forms a tetramer. I think it important, however, that the authors are more transparent in the main manuscript about the ambiguity in the SEC-MALS data in Fig S1. They should also reword the following sentence at the beginning of this section, which in light of the SEC-MALS data is misleading: “The AdoCbl-bound TtCBD showed identical light-activated spectroscopic behaviour to the full-length TtCarH protein.”

We have altered this to “The AdoCbl-bound TtCBD showed similar light-activated spectroscopic behaviour to the full-length TtCarH protein.”

In contrast to ref 12, who apparently show using SEC and UV-visible absorption that under illumination the full length H132A 4mers convert mostly to 1mers, in Fig S13 the authors find only partial monomerization of this variant under illumination using similar measures (plus native mass spec in Fig S2). Could this partial photoconversion be owing to differences in illumination conditions (intensity, illumination time etc)? This should be discussed for the avoidance of doubt.

We have now included: we find that the majority of the aerobic light exposed TtCarH H132A (both full-length and CBD-only) remain in the tetrameric state (Figure S1, Figure S13a-d) in comparison to the Wt protein under identical illumination conditions.

Also, their evidence in Fig S14 that the light-adapted TtCBD-H132A has a spectrum more consistent with a hydroxylated Co(III) species than the bis-His adduct is convincing and an interesting finding. It's not clear to me from the data, however, if this species is formed exclusively in the truncated CBD protein or also in this variant for the full-length protein?

We have altered the sentence to make clear this is previously observed for full-length H132A as well as in our own data (Fig S13d). Our manuscript now reads “The absorbance spectrum for the dark state of TtCBD-H132A resembles that of wild type TtCBD, whereas the light-adapted spectrum shows an absorbance peak at 352 nm instead of 359 nm, indicating formation of a hydroxylated Co(III) species instead of the bis-His ligated state also reported previously for the full-length H132A variant¹² (Figure S13d, Figure S14a and b).”

In response to my review, the authors added the following passage to the introduction: “Kutta et al suggested that the Co-C bond scission is likely to proceed via a heterolytic cleavage route, leading to further distinct spectral intermediates on the μ s-ms timescales and tetramer dissociation over several seconds²¹. However, Miller and co-workers have proposed the formation of a long-lived triplet state upon excitation, which ultimately reacts to form the stable 4', 5' anhydroadenosine product and final light state of CarH²².” As written, this sounds like heterolytic bond breakage and a long-lived triplet state a mutually exclusive, which they aren't. They should make the case for why they think they are or edit this accordingly to remove this implication.

We have edited this to: Kutta et al suggested that the Co-C bond scission is likely to proceed via a heterolytic cleavage route, leading to further distinct spectral intermediates on the μ s-ms timescales and tetramer dissociation over several seconds²¹. It has also been proposed a long-lived triplet state is formed upon excitation, which ultimately reacts to form the stable 4', 5' anhydroadenosine product and final light state of CarH²².